# A Token is Worth over 1,000 Tokens: Efficient Knowledge Distillation through Low-Rank Clone

**Jitai Hao**[1,*]   **Qiang Huang**[1,†]   **Hao Liu**[2]   **Xinyan Xiao**[2]   **Zhaochun Ren**[3]   **Jun Yu**[1,4,†]

[1]School of Intelligence Science and Engineering, Harbin Institute of Technology, Shenzhen
[2]Baidu Inc.   [3]Leiden University   [4]Pengcheng Laboratory
jitaihao@stu.hit.edu.cn, {huangqiang, yujun}@hit.edu.cn
{liuhao24, xiaoxinyan}@baidu.com, z.ren@liacs.leidenuniv.nl

## Abstract

Training high-performing Small Language Models (SLMs) remains computationally expensive, even with knowledge distillation and pruning from larger teacher models. Existing approaches often face three key challenges: (1) information loss from hard pruning, (2) inefficient alignment of representations, and (3) underutilization of informative activations, particularly from Feed-Forward Networks (FFNs). To address these challenges, we introduce **Low-Rank Clone (LRC)**, an efficient pre-training method that constructs SLMs aspiring to behavioral equivalence with strong teacher models. LRC trains a set of low-rank projection matrices that jointly enable soft pruning by compressing teacher weights, and activation clone by aligning student activations, including FFN signals, with those of the teacher. This unified design maximizes knowledge transfer while removing the need for explicit alignment modules. Extensive experiments with open-source teachers such as Llama-3.2-3B-Instruct and Qwen2.5-3B/7B-Instruct show that LRC matches or surpasses the performance of state-of-the-art models trained on trillions of tokens–using only 20B tokens, achieving over **1,000×** greater training efficiency. Our codes and model checkpoints are available at Github and Huggingface.

## 1   Introduction

Large Language Models (LLMs) have shown exceptional performance across a wide range of Natural Language Processing (NLP) tasks [2, 74, 73, 22, 25]. However, their deployment remains limited in real-world applications due to their immense computational and memory requirements, making them unsuitable for latency-sensitive, edge-based, or privacy-preserving scenarios. As a result, there is growing momentum toward developing Small Language Models (SLMs) that offer similar capabilities with significantly lower resource footprints.

Yet, despite their smaller size, training high-performing SLMs remains a resource-intensive endeavor. For example, state-of-the-art SLMs such as Llama-3.2-3B [4] and Qwen3-1.7B [74] require 9

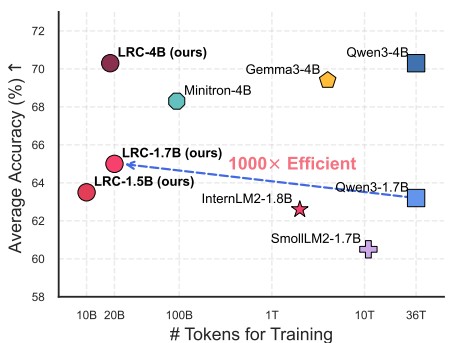

Figure 1: LRC results that achieve higher accuracy with 1,000× fewer training tokens, significantly boosting efficiency.

---

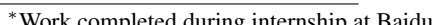

*Work completed during internship at Baidu.

†Corresponding authors.

39th Conference on Neural Information Processing Systems (NeurIPS 2025).

and 36 trillion tokens, respectively, for pre-training. To reduce such substantial costs, knowledge distillation [31] has emerged as a key strategy, enabling a compact student model to learn from a larger and more powerful teacher model [41, 53, 23, 22]. Recent efforts such as Minitron [53] and Sheared Llama [70] have explored the use of existing LLMs [74, 1, 22, 3] to accelerate SLM pre-training. These methods typically combine structured pruning and distillation, first removing "unimportant" neurons and then recovering performance via distillation or continued training.

Despite these advances, current distillation paradigms still fall short in fully utilizing the rich knowledge embedded in teacher models, resulting in limited efficiency and suboptimal student performance. We identify three core challenges in existing approaches:

- **Information Loss from Hard Pruning:** Most existing methods use *hard pruning*, permanently removing selected neurons, channels, attention heads, or entire layers [75, 53, 70, 51]. While reducing model size, it discards valuable information from the teacher's weights, e.g., pruning 50% of Llama-7B in LLM-Pruner [48] caused a sharp performance drop from 63.25 to 48.98.
- **Inefficient Alignment of Representations:** Feature-based distillation methods [37, 67, 53] often use additional projection matrices to align intermediate activations between teacher and student. However, as the student's internal states evolve during training, learning effective alignment mappings becomes challenging, reducing distillation efficiency.
- **Underutilization of Informative Activations:** Prior work has primarily focused on aligning attention scores [37, 67], while largely overlooking the high-dimensional, information-rich activations from Feed-Forward Networks (FFNs). These FFN signals are crucial to the expressiveness of modern LLMs, as confirmed by our ablation study in Section 4.3.

To overcome these challenges, we propose **Low-Rank Clone (LRC)**, a highly efficient pre-training method that constructs SLMs aspiring to behavioral equivalence with strong teacher models. LRC eliminates the need to train the student's weights, except for the RMSNorm parameters [76], which constitute less than 1% of the total, drastically reducing training overhead.

Compared to prior multi-stage approaches [53, 70], LRC introduces a unified framework that performs *soft pruning* and knowledge distillation simultaneously via trainable low-rank projection matrices. Each forward pass of LRC consists of two key steps: (1) **Low-Rank Projection**, which projects the teacher's weights into smaller weights that directly serve as the student's parameters. (2) **Activation Clone**, which aligns the student's intermediate activations with those of the teacher to preserve behavioral fidelity. This design directly addresses the core limitations of existing methods, offering three key advantages:

- **Minimal Information Loss:** By directly generating the student's weights from the teacher model, LRC preserves substantially more of the teacher's knowledge, even at aggressive compression levels, than hard pruning strategies.
- **Alignment-Free Distillation:** The projection matrices naturally handle representational mismatches between teacher and student layers, removing the need for additional alignment modules and improving both efficiency and performance.
- **Full Utilization of Activations:** LRC captures a broad spectrum of intermediate signals, including underutilized FFN activations, which we show to encode rich and valuable information often ignored by prior methods.

We comprehensively evaluate LRC using strong, open-source teacher models such as Llama-3.2-3B-Instruct and Qwen2.5-3B/7B-Instruct, showing competitive or superior performance compared to leading SLMs trained on trillions of tokens. As depicted in Figure 1, LRC-1.7B outperforms Qwen3-1.7B (64.98 vs. 63.17) on standard benchmarks, while requiring up to **1,000× fewer** training tokens. These results highlight LRC's potential to drastically improve the cost-efficiency of high-performance SLM development and advance the state of practical, high-performance language models.

## 2 Related Work

To reduce the computational and memory overhead of LLMs, researchers have explored a range of techniques, including knowledge distillation [57, 62, 31, 37, 24, 67, 23], structured pruning [15, 75, 61, 48, 51], quantization [16, 71, 45], KV cache compression [72, 27, 36, 77, 47], and efficient training frameworks [6, 60, 78, 26, 35]. Among these, knowledge distillation and structured pruning are most relevant to our work.

**Knowledge Distillation and Structured Pruning.** Knowledge distillation [31, 38] aims to transfer knowledge from a large, pre-trained teacher model to a smaller student model. Early approaches either use synthetic data generated by the teacher to train the student [11, 55, 32], or minimize the divergence between their output distributions [31, 24, 59]. While effective, these techniques often suffer from limited transfer efficiency and scalability [23, 57].

To improve transfer quality, feature-based distillation methods like TinyBert [37], MiniLM [67] and TED [42] utilize intermediate activations from transformer layers to guide student learning. However, these methods ignore the rich information encoded in the model weights [21], and require additional alignment matrices to bridge discrepancies in hidden states, increasing training overhead. In contrast, LRC circumvents these limitations by using trainable low-rank projection matrices that simultaneously extract weight-level knowledge and serve as implicit alignment layers, eliminating the need for student weight initialization or separate alignment training.

Recent approaches such as Minitron [53] and Sheared Llama [70] integrate hard pruning with distillation to compress LLMs. Yet, they rely on a multi-stage pipeline–pruning followed by distillation or continued training–which increases training cost and sensitivity to pruning ratios. Moreover, hard pruning can cause substantial performance degradation [48]. By contrast, LRC performs soft pruning and distillation in a simple single stage, improving both efficiency and model performance.

Our model-centric method is complementary to recent data-centric approaches. For example, LIMA [80] has shown that high-quality data is crucial for the alignment phase, while DA-KD [29] introduces a framework for data filtering based on difficulty.

Structured pruning remains a key technique for LLM compression [48, 61, 19, 70]. A recent example, SliceGPT [7], uses orthogonal projection and PCA [50] to prune weights while maintaining computational equivalence. Nevertheless, PCA's linear assumptions often fail to capture the nonlinear nature of LLM weights, limiting its performance and compression capacity. Instead, LRC adopts learnable low-rank projections that better adapt to the underlying structure of transformer weights, improving both compression fidelity and knowledge retention.

**Small Language Models (SLMs).** SLMs have emerged as a practical solution for deploying language models in resource-constrained environments. Recent efforts aim to train SLMs that approach LLM-level performance [5, 74, 46, 34]. Nonetheless, even with the help of distillation [22], achieving strong performance still typically requires pre-training on tens of trillions of tokens [4], limiting accessibility and practicality. Unlike prior methods, LRC achieves competitive performance with only **10 billion** tokens, offering a paradigm shift in the efficiency of SLM training.

## 3 Low-Rank Clone

We present **Low-Rank Clone (LRC)**, a novel distillation method that aims to construct SLMs approaching behavioral equivalence with strong teacher models. As illustrated in Figure 2, LRC consists of two key steps: (1) **Low-Rank Projection** that compresses the teacher's weights into a compact space, and (2) **Activation Clone** that aligns the activations of the student with those of the teacher to preserve behavioral fidelity during forward passes.

**Background and Notation.** LRC builds on the transformer architecture as used in models like Llama [66, 22]. Each transformer layer mainly consists of a self-attention mechanism and an FFN. The attention mechanism involves four weight matrices: $\boldsymbol{W}_q \in \mathbb{R}^{d_\mathrm{q} \times d}$, $\boldsymbol{W}_\mathrm{k} \in \mathbb{R}^{d_\mathrm{kv} \times d}$, $\boldsymbol{W}_\mathrm{v} \in \mathbb{R}^{d_\mathrm{kv} \times d}$, and $\boldsymbol{W}_\mathrm{o} \in \mathbb{R}^{d \times d}$, where $d$ is the hidden size of the model, and $d_\mathrm{q}, d_\mathrm{kv}$ denote the query/key/value dimensions. Given an input vector $\boldsymbol{x} \in \mathbb{R}^d$, the attention output is:

$$\boldsymbol{o}_\mathrm{attn} = \mathrm{Attn}(\boldsymbol{x}\boldsymbol{W}_q^\top, \boldsymbol{x}\boldsymbol{W}_\mathrm{k}^\top, \boldsymbol{x}\boldsymbol{W}_\mathrm{v}^\top)\boldsymbol{W}_\mathrm{o}.$$

The FFN employs the SwiGLU activation [58, 66, 74], containing three weight matrices, i.e., $\boldsymbol{W}_\mathrm{up} \in \mathbb{R}^{d_\mathrm{mid} \times d}$, $\boldsymbol{W}_\mathrm{gate} \in \mathbb{R}^{d_\mathrm{mid} \times d}$, and $\boldsymbol{W}_\mathrm{down} \in \mathbb{R}^{d_\mathrm{mid} \times d}$, where $d_\mathrm{mid}$ represents the intermediate dimension. The computation of the SwiGLU-based FFN is defined as:

$$\boldsymbol{o}_\mathrm{ffn} = \mathrm{SwiGLU}(\boldsymbol{x}\boldsymbol{W}_\mathrm{up}^\top, \boldsymbol{x}\boldsymbol{W}_\mathrm{gate}^\top)\boldsymbol{W}_\mathrm{down},$$

where $\mathrm{SwiGLU}(\boldsymbol{x}, \boldsymbol{y}) = \boldsymbol{x} \odot \sigma(\boldsymbol{y})$, with $\sigma$ being the SiLU activation function, and $\odot$ denoting element-wise multiplication. RMSNorm [76] is typically the normalization technique after both the

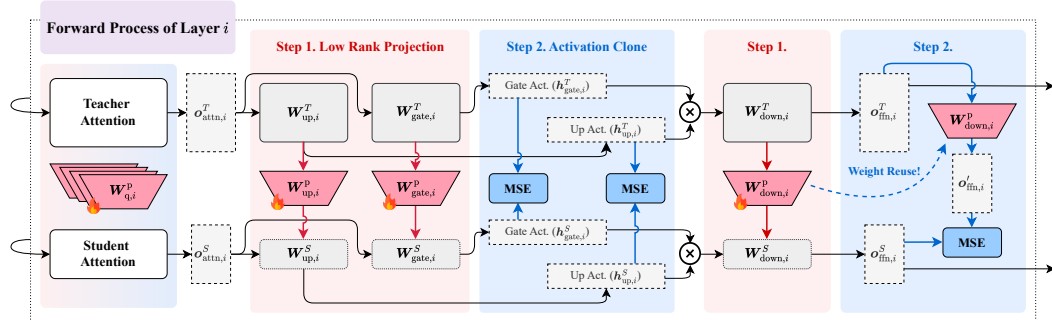

Figure 2: The overall procedure of LRC. To ensure clarity, attention and normalization modules are omitted. LRC involves two main steps: (1) Low-Rank Projection: applying low-rank projection matrices to compress the teacher's weights into a lower-dimensional space, which are then assigned to the student. (2) Activation Clone, executing standard forward passes in both models to collect intermediate activations, which are aligned using Mean Squared Error (MSE) loss.

attention and FFN components, which is defined as:

$$\text{RMSNorm}(\boldsymbol{x}) = \frac{\boldsymbol{x}}{\sqrt{\frac{1}{d}\sum_{i=1}^{d} x_i^2 + \epsilon}} \odot \boldsymbol{g},$$

where $\boldsymbol{g} \in \mathbb{R}^d$ is a learnable scaling parameter and $\epsilon$ is a small constant added for numerical stability. Given a vocabulary $V$, the embedding matrix $\boldsymbol{W}_{\text{emb}} \in \mathbb{R}^{|V| \times d}$ transforms input token indices into embeddings of dimension $d$. At the output, the language model (LM) head $\boldsymbol{W}_{\text{lm}} \in \mathbb{R}^{|V| \times d}$ projects the final hidden states back into vocabulary logits. In SLMs, the LM head usually shares weights with the embedding matrix [22, 74], reducing parameter redundancy.

## 3.1 Low-Rank Projection

Conventional feature-based distillation methods typically initialize student weights either from scratch [37] or by pruning subsets of the teacher model [70, 53, 42, 57]. While straightforward, these approaches inevitably discard valuable information and suffer from reduced distillation efficiency.

To address this, LRC introduces a Low-Rank Projection step that replaces manual initialization with a principled, trainable transformation. As shown in Figure 2, a set of low-rank projection matrices:

$$\boldsymbol{W}_{m,i}^{\text{P}}, \boldsymbol{W}_{\text{emb}}^{\text{P}}, \boldsymbol{W}_{\text{lm}}^{\text{P}}, \quad m \in \{\text{q}, \text{k}, \text{v}, \text{o}, \text{up}, \text{gate}, \text{down}\}, \quad 0 < i < l,$$

are used to map the teacher's weights into a lower-dimensional student space, where $l$ is the number of layers. These matrices, together with the student's RMSNorm parameters [76], constitute the *only trainable components* in LRC. Since RMSNorm contributes less than 1% of the total trainable parameters, we focus below on the projection process in two stages.

**Attention and FFN Weight Projection.** For each layer $i$, LRC generates the student's attention and FFN weights by applying low-rank projections to the corresponding teacher weights:

$$\boldsymbol{W}_{m,i}^{\text{S}} = \boldsymbol{W}_{m,i}^{\text{T}} \boldsymbol{W}_{m,i}^{\text{P}}, \tag{1}$$

where $m \in \{\text{q}, \text{k}, \text{v}, \text{o}, \text{up}, \text{gate}, \text{down}\}$, $\boldsymbol{W}_{m,i}^{\text{T}} \in \mathbb{R}^{d_m^{\text{T}} \times d^{\text{T}}}$, $\boldsymbol{W}_{m,i}^{\text{P}} \in \mathbb{R}^{d^{\text{T}} \times d^{\text{S}}}$, and $\boldsymbol{W}_{m,i}^{\text{S}} \in \mathbb{R}^{d_m^{\text{T}} \times d^{\text{S}}}$. Here, the hidden sizes follow: (1) $d_{\text{o}} = d$, (2) $d_{\text{k}} = d_{\text{v}} = d_{\text{kv}}$, and (3) $d_{\text{gate}} = d_{\text{up}} = d_{\text{down}} = d_{\text{mid}}$. Superscripts T and S refer to teacher and student, respectively.

**Embedding and LM Head Projection.** The embedding and LM head weights are projected in the same manner:

$$\boldsymbol{W}_m^{\text{S}} = \boldsymbol{W}_m^{\text{T}} \boldsymbol{W}_{\text{emb}}^{\text{P}}, \tag{2}$$

where $m \in \{\text{emb}, \text{lm}\}$, $\boldsymbol{W}_m^{\text{T}} \in \mathbb{R}^{|V| \times d^{\text{T}}}$, $\boldsymbol{W}_m^{\text{P}} \in \mathbb{R}^{d^{\text{T}} \times d^{\text{S}}}$, and $\boldsymbol{W}_m^{\text{S}} \in \mathbb{R}^{|V| \times d^{\text{S}}}$. Here, $V$ is the shared vocabulary of the teacher and student models.

**Structural Compatibility and Deployment.** The resulting student model retains full architectural compatibility and can perform forward passes without modification. Importantly, this enables immediate post-training and inference of the student model.

**Algorithm 1:** Overall Procedure of LRC

---

**Input:** Input token sequence $\mathcal{T}$; number of layers $l$; RMSNorm constant $\epsilon$; teacher's weights $\{\boldsymbol{W}_{m,i}^{\mathrm{T}}\}, \boldsymbol{W}_{\mathrm{emb}}^{\mathrm{T}}, \boldsymbol{W}_{\mathrm{lm}}^{\mathrm{T}}$; low-rank projection matrices $\{\boldsymbol{W}_{m,i}^{\mathrm{P}}\}, \boldsymbol{W}_{\mathrm{emb}}^{\mathrm{P}}, \boldsymbol{W}_{\mathrm{lm}}^{\mathrm{P}}$;

**Output:** Clone loss $\mathcal{L}_{\mathrm{clone}}$;

▷ Step 1: Low-Rank Projection

1 **for** $i = 1$ **to** $l$ **do**

2     **foreach** $m \in \{\mathrm{q, k, v, o, up, gate, down}\}$ **do**

3        $\boldsymbol{W}_{m,i}^{\mathrm{S}} \leftarrow \boldsymbol{W}_{m,i}^{\mathrm{T}} \boldsymbol{W}_{m,i}^{\mathrm{P}}$;                 ▷ Generate student weights

4 $\boldsymbol{W}_{\mathrm{emb}}^{\mathrm{S}} \leftarrow \boldsymbol{W}_{\mathrm{emb}}^{\mathrm{T}} \boldsymbol{W}_{\mathrm{emb}}^{\mathrm{P}}; \boldsymbol{W}_{\mathrm{lm}}^{\mathrm{S}} \leftarrow \boldsymbol{W}_{\mathrm{lm}}^{\mathrm{T}} \boldsymbol{W}_{\mathrm{lm}}^{\mathrm{P}}$;

▷ Step 2: Activation Clone

5 $\mathcal{L}_{\mathrm{clone}} \leftarrow 0$;

6 $\boldsymbol{h}^{\mathrm{T}}, \boldsymbol{o}_{\mathrm{attn}}^{\mathrm{T}}, \boldsymbol{o}_{\mathrm{ffn}}^{\mathrm{T}} \leftarrow \texttt{Forward}(\mathcal{T}, l, \epsilon, \{\boldsymbol{W}_{m,i}^{\mathrm{T}}\}, \boldsymbol{W}_{\mathrm{emb}}^{\mathrm{T}}, \boldsymbol{W}_{\mathrm{lm}}^{\mathrm{T}})$;    ▷ Get teacher act. dict.

7 $\boldsymbol{h}^{\mathrm{S}}, \boldsymbol{o}_{\mathrm{attn}}^{\mathrm{S}}, \boldsymbol{o}_{\mathrm{ffn}}^{\mathrm{S}} \leftarrow \texttt{Forward}(\mathcal{T}, l, \epsilon, \{\boldsymbol{W}_{m,i}^{\mathrm{S}}\}, \boldsymbol{W}_{\mathrm{emb}}^{\mathrm{S}}, \boldsymbol{W}_{\mathrm{lm}}^{\mathrm{S}})$;    ▷ Get student act. dict.

8 **for** $i = 1$ **to** $l$ **do**

9     **foreach** $m \in \{\mathrm{q, k, v, gate, up}\}$ **do**     ▷ Compute clone loss of interm. states

10        $\mathcal{L}_{\mathrm{clone}} \leftarrow \mathcal{L}_{\mathrm{clone}} + \mathcal{E}(\boldsymbol{h}_{m,i}^{\mathrm{S}}, \boldsymbol{h}_{m,i}^{\mathrm{T}})$;

11     $\mathcal{L}_{\mathrm{clone}} \leftarrow \mathcal{L}_{\mathrm{clone}} + \mathcal{E}(\boldsymbol{o}_{\mathrm{attn},i}^{\mathrm{S}}, \boldsymbol{o}_{\mathrm{attn},i}^{\mathrm{T}} \boldsymbol{W}_{\mathrm{o},i}^{\mathrm{P}}) + \mathcal{E}(\boldsymbol{o}_{\mathrm{ffn},i}^{\mathrm{S}}, \boldsymbol{o}_{\mathrm{ffn},i}^{\mathrm{T}} \boldsymbol{W}_{\mathrm{down},i}^{\mathrm{P}})$;

12 **return** $\mathcal{L}_{\mathrm{clone}}$;

---

## 3.2 Activation Clone

While previous methods have leveraged attention states to improve distillation efficiency [67, 37], they largely overlook the rich information contained in FFN activations. To capture more comprehensive semantic signals, LRC aligns a wide range of intermediate activations, treating them as fine-grained *anchor points* for behavioral replication.

Specifically, LRC matches both the internal linear projections $\boldsymbol{h}_m = \boldsymbol{x} \boldsymbol{W}_m^\top$, where $m \in \{\mathrm{q, k, v, up, gate}\}$, and the output vectors $\boldsymbol{o}_{\mathrm{attn}}$ and $\boldsymbol{o}_{\mathrm{ffn}}$ from the attention and FFN modules, respectively. As depicted in Figure 2, all these activations are aligned using the Mean Squared Error (MSE) loss $\mathcal{E}$, yielding the overall Activation Clone loss $\mathcal{L}_{\mathrm{clone}}$:

$$\mathcal{L}_{\mathrm{clone}} = \sum_i^l \left[ \mathcal{E}(\boldsymbol{o}_{\mathrm{attn},i}^{\mathrm{S}}, \boldsymbol{o}_{\mathrm{attn},i}^{\mathrm{T}} \boldsymbol{W}_{\mathrm{o},i}^{\mathrm{P}}) + \mathcal{E}(\boldsymbol{o}_{\mathrm{ffn},i}^{\mathrm{S}}, \boldsymbol{o}_{\mathrm{ffn},i}^{\mathrm{T}} \boldsymbol{W}_{\mathrm{down},i}^{\mathrm{P}}) + \sum_m \mathcal{E}(\boldsymbol{h}_{m,i}^{\mathrm{S}}, \boldsymbol{h}_{m,i}^{\mathrm{T}}) \right], \quad (3)$$

where $m \in \{\mathrm{q, k, v, up, gate}\}$. Following prior work [22, 24], LRC also employs a KL divergence loss $\mathcal{L}_{\mathrm{KL}}$ to align teacher and student logits over the vocabulary and a next-token prediction loss $\mathcal{L}_{\mathrm{LM}}$ to enhance model performance. The total training objective is:

$$\mathcal{L} = \mathcal{L}_{\mathrm{KL}} + \mathcal{L}_{\mathrm{LM}} + \alpha \mathcal{L}_{\mathrm{clone}}, \quad (4)$$

where $\alpha$ is a hyperparameter controlling the weight of activation alignment.

**Alignment Free Design.** Vanilla feature-based distillation approaches require additional projection matrices to reconcile mismatched hidden dimensions [37, 42]. In contrast, LRC is inherently *alignment-free*, i.e., the same low-rank projection matrices used to generate the student's weights (e.g., $\boldsymbol{W}_{\mathrm{o}}, \boldsymbol{W}_{\mathrm{down}}$) can be reused directly to align activations during training. This property arises from the structure of transformer modules, where outputs are linear combinations of their respective output projection weights. Here, we illustrate this using the FFN. Formally, we have Lemma 1 as follows:

**Lemma 1** (Alignment-Free FFN Output Cloning). *Let $\boldsymbol{W}_{\mathrm{down},i}^{\mathrm{S}}$ denote the FFN down-projection weight in the student model at layer $i$, derived via the low-rank projection from the teacher's weight $\boldsymbol{W}_{\mathrm{down},i}^{\mathrm{T}}$ and projection matrix $\boldsymbol{W}_{\mathrm{down},i}^{\mathrm{P}}$, such that:*

$$\boldsymbol{W}_{\mathrm{down},i}^{\mathrm{S}} = \boldsymbol{W}_{\mathrm{down},i}^{\mathrm{T}} \boldsymbol{W}_{\mathrm{down},i}^{\mathrm{P}}.$$

*If the intermediate FFN activations $\boldsymbol{h}_{\mathrm{up},i}$ and $\boldsymbol{h}_{\mathrm{gate},i}$ are perfectly cloned, i.e.,*

$$\boldsymbol{h}_{\mathrm{up},i}^{\mathrm{S}} = \boldsymbol{h}_{\mathrm{up},i}^{\mathrm{T}}, \ \ \boldsymbol{h}_{\mathrm{gate},i}^{\mathrm{S}} = \boldsymbol{h}_{\mathrm{gate},i}^{\mathrm{T}},$$

Table 1: Zero-shot performance comparison between LRC and state-of-the-art publicly available models with fewer than 2B parameters. "# Tokens" denotes the number of training tokens; "N/A" indicates unavailable training data. All models, including teachers and ours, are instruct versions.

| Model | InternLM2-1.8B | **LRC-1.7B** | Qwen3-1.7B | SmolLM2-1.7B | **LRC-1.5B** | MiniCPM-1.2B |
|---|---|---|---|---|---|---|
| **Teacher** | – | Qwen2.5-3B | – | – | Llama3-3B | – |
| **# Tokens** | 2T | **20B** | 36T | 11T | **10B** | 1T |
| **Dataset** | N/A | Mixed-1.1 | N/A | SomlLM | Mixed-1.1 | N/A |
| **ARC-E** | 71.04 | 74.62 | 72.47 | 69.11 | 74.75 | 70.16 |
| **ARC-C** | 42.06 | 44.20 | 43.00 | 43.52 | 44.97 | 39.68 |
| **LogiQA** | 28.42 | 30.88 | 28.42 | 28.88 | 30.72 | 30.88 |
| **CSQA** | 70.11 | 70.19 | 64.78 | 51.19 | 65.77 | 64.29 |
| **PIQA** | 74.27 | 73.07 | 72.20 | 76.01 | 73.07 | 74.65 |
| **WinoG** | 63.77 | 63.30 | 61.48 | 68.98 | 62.25 | 60.77 |
| **BoolQ** | 75.50 | 79.82 | 77.65 | 68.47 | 75.78 | 67.58 |
| **SciQ** | 94.50 | 93.80 | 93.10 | 89.80 | 94.60 | 91.50 |
| **MMLU** | 43.75 | 54.93 | 55.44 | 48.50 | 49.42 | 44.23 |
| **Avg. ↑** | 62.60 | **64.98** | 63.17 | 60.50 | **63.48** | 60.42 |

*then the student FFN output is exactly equal to the teacher output passed through the same projection:*

$$\mathcal{E}(\boldsymbol{o}_{\mathrm{ffn},i}^{\mathrm{S}}, \boldsymbol{o}_{\mathrm{ffn},i}^{\mathrm{T}} \boldsymbol{W}_{\mathrm{down},i}^{\mathrm{P}}) = 0.$$

The proof is provided in Appendix A. Lemma 1 shows that LRC needs no additional alignment matrices–its low-rank projections suffice for both weight transformation and activation alignment.

**Remarks.** The overall procedure of LRC is summarized in Algorithm 1. The `Forward` function executes a standard transformer forward pass and collects intermediate activations $\boldsymbol{h}_{m,i}$ (for $m \in \{q, k, v, up, gate\}$) and the outputs $\boldsymbol{o}_{\mathrm{attn},i}$ and $\boldsymbol{o}_{\mathrm{ffn},i}$ of the attention and FFN modules at each layer. Pseudo-code for this function is provided in Appendix C.

## 4 Experiments

### 4.1 Experiment Setup

**Training Configuration.** We train a series of LRC models using strong open-source SLMs as teachers, i.e., Llama-3.2-3B-Instruct [22] for LRC-1.5B, Qwen2.5-3B-Instruct [74] for LRC-1.7B, and Qwen2.5-7B-Instruct for LRC-4B. To fairly compare with Sheared-Llama [70], we also train LRC-2.7B using Llama-2-7B-chat as the teacher. We employ supervised fine-tuning (SFT) to obtain the instructed versions of LRC models. Implementation details are provided in Appendices D.1 and D.2. All models are trained with packed sequences of length 2,048 for computational efficiency. We use the Adam optimizer with $\beta_1 = 0.9$ and $\beta_2 = 0.999$, and set the KL divergence temperature to 40. Training runs on 8 NVIDIA H800 GPUs using `PyTorch`, `transformers` [69], and `deepspeed` [6] for distributed parallelism. The hyperparameter settings and model configurations are provided in Appendices D.3 and D.4, respectively.

**Training Datasets.** We construct the training corpus by mixing data from Fineweb-Edu [54], DCLM [40], and CosmopiediaV2 [5]. Fineweb-Edu served as the primary component, selected for its high-quality educational content. To enrich the pre-training data distribution, we incorporate DCLM and CosmopiediaV2, and use OpenHermes [65]. We also utilize UltraChat [18] as a supervised fine-tuning dataset for instruction-tuning. The combined pre-training dataset is randomly shuffled without curriculum settings. Data composition ratios and sources are listed in Appendix D.5.

**Baselines.** We compare LRC against several representative and competitive baselines: (1) Sheared Llama [70], using the same teacher and training data for a fair comparison; (2) Minitron [53], evaluated via its released checkpoint; (3) TinyBERT [37], a feature-based distillation method adapted to the Llama architecture. We also benchmark LRC against state-of-the-art open-source SLMs of similar sizes, including MiniCPM [34], SmolLM2 [5], Gemma3 [64], InternLM [10], and models from the Qwen3 [73] families. Model checkpoints are listed in Appendix D.2.

**Evaluation Protocol.** In the experiments, all models are evaluated in zero-shot settings using the `lm-evaluation-harness` framework [20], with `transformers` [69] serving as the inference back-

Table 2: Zero-shot performance comparison between LRC and state-of-the-art publicly available models with more than 2B parameters, where the model with "-B" refers to pre-trained only.

| Model | Gemma3-4B | Minitron-4B | Qwen3-4B | LRC-4B | LRC-2.7B-B | Sheared-Llama-2.7B-B |
|---|---|---|---|---|---|---|
| **Teacher** | – | Nemotron4-15B | – | Qwen2.5-7B | Llama2-7B | Llama2-7B |
| **# Tokens** | 4T | 94B | 36T | **18B** | **10B** | 50B |
| **Dataset** | N/A | N/A | N/A | Mixed-2.0 | Redpajama | Redpajama |
| **ARC-E** | 82.53 | 79.59 | 80.47 | 78.37 | 58.59 | 67.30 |
| **ARC-C** | 57.08 | 54.35 | 53.58 | 52.47 | 29.61 | 33.58 |
| **LogiQA** | 33.03 | 30.26 | 33.64 | 34.10 | 29.03 | 28.26 |
| **CSQA** | 69.37 | 71.09 | 75.76 | 79.28 | 36.36 | 18.92 |
| **PIQA** | 76.44 | 77.64 | 75.08 | 76.82 | 66.97 | 76.17 |
| **WinoG** | 69.38 | 65.93 | 65.27 | 67.72 | 62.43 | 65.04 |
| **BoolQ** | 83.94 | 82.60 | 84.95 | 84.50 | 74.31 | 65.99 |
| **SciQ** | 95.50 | 96.60 | 95.50 | 95.00 | 85.50 | 91.10 |
| **MMLU** | 57.58 | 56.77 | 68.38 | 64.41 | 31.20 | 26.56 |
| **Avg. ↑** | 69.43 | 68.31 | 70.29 | **70.30** | **52.67** | 52.55 |

end. We assess performance across a suite of downstream tasks covering a range of language understanding skills: (1) Scientific and Logical Reasoning (ARC-E [13], ARC-C [13], and LogiQA [44]), (2) Commonsense Understanding (CommonsenseQA (CSQA) [63], PIQA [9], and WinoGrande (WinoG) [56]), (3) Reading Comprehension (BoolQ [12]), and (4) World Knowledge (SciQ [68] and MMLU [30]). Downstream task and evaluation metric details are provided in Appendix D.6.

## 4.2 Main Results

We begin by comparing LRC models with fewer than 2B parameters against leading SLMs, as shown in Table 1. LRC-1.5B, distilled from Llama-3.2-3B-Instruct using only 10B tokens, outperforms SmolLM2-1.7B, which was trained on 11T tokens. Similarly, LRC-1.7B, trained from Qwen2.5-3B-Instruct, achieves the best performance among all models below 2B, surpassing Qwen3-1.7B, which was trained on 36T tokens. These results highlight LRC's remarkable distillation efficiency, achieving superior performance with more than **1000× fewer** training tokens.

To assess scalability, we further evaluate LRC on larger models in Table 2. LRC-4B, distilled from Qwen2.5-7B-Instruct using just 10B tokens, achieves performance comparable to Qwen3-4B (trained on 36T tokens), and outperforms Minitron-4B, which was trained with 5× more data. We also conduct a fair comparison with Sheared-Llama-2.7B-B by replicating its setup using Llama2-7B as the teacher and an identical training dataset without dynamic batch loading [70] for improved data quality. Our LRC-2.7B-B still achieves comparable performance while using 5× fewer tokens. Here, "-B" indicates pre-training only (i.e., no SFT).

These findings demonstrate LRC's robustness and generality across diverse teacher-student configurations. Notably, all reported LRC models are followed by SFT. We further analyze the impact of SFT in Appendix E.1. Additionally, we evaluate the LRC performance on few-shot tasks, where the results and analysis are provided in Appendix E.2.

## 4.3 Ablation Study

We conduct an ablation study to assess the contributions of LRC's two core components: **Low-Rank Projection** and **Activation Clone**. All experiments use Llama-3.2-3B-Instruct as the teacher and are trained on 2.5B tokens without performing SFT. We report training LM loss as the evaluation metric, as the data contains minimal duplication and training runs for only one epoch.

**Low-Rank Projection.** To assess the impact of low-rank projection, we compare against TinyBERT-style distillation, where the student is randomly initialized and trained from scratch using MSE loss with attention activations and outputs of each layer. We implement TinyBERT for the Llama architecture. As it relies on attention score maps, TinyBERT struggles to scale to longer contexts since it cannot use FlashAttention [14]. The adaptations are detailed in Appendix D.7. As shown in Figure 3, LRC reaches an LM loss of 3.0 nearly 2.7× faster than TinyBERT, highlighting the benefit of transferring structured weight information through projection rather than learning from scratch.

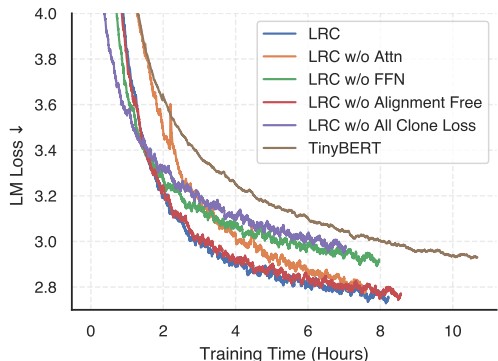

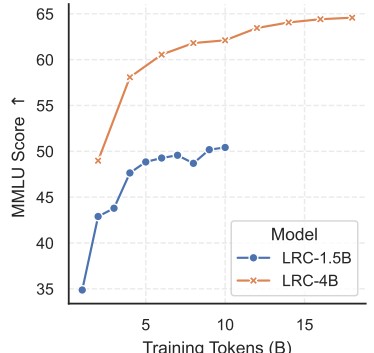

Figure 3: Effect of LRC component ablations on LM loss convergence over training time.

Figure 4: The trend of MMLU scores with increasing training tokens.

Table 3: Ablation results for removing different terms of the clone loss. Scores significantly higher than the baseline (None, with all losses retained) are underline.

| Removed Term | None | Attn q | Attn k | Attn v | Attn o | FFN gate | FFN up | FFN down |
|---|---|---|---|---|---|---|---|---|
| LM Loss ↓ | 2.639 | 2.630 | 2.629 | 2.639 | 2.636 | 2.677 | 2.639 | 2.651 |

**Activation Clone.** To measure the contribution of different activation signals in the clone loss $\mathcal{L}_{\mathrm{clone}}$, we conduct both term-level and module-level ablations. Details are provided in Appendix D.8. We also test layer-level ablations, and the results are shown in Appendix E.6.

Table 3 presents the term-level results when individual activation terms are removed. Removing FFN-related terms, particularly FFN gate, significantly degrades performance, increasing LM loss from 2.639 to 2.677. This confirms that FFN activations carry essential information and that aligning them is crucial for effective behavioral cloning.

Figure 3 depicts the module-level results, where we show the impact of dropping all attention-related vs. FFN-related clone losses, as well as removing all clone signals entirely. We observe that LRC w/o Attn, while significantly impacting performance in the early training stages, gradually recovers and converges toward the performance of full LRC in later stages. However, LRC w/o FFN produces a substantial performance degradation that persists throughout training, further confirming the critical importance of FFN activations. In addition, when both LRC and LRC w/o All Clone Loss reach an LM loss of 3.0, LRC achieves more than $2\times$ reduction in training time usage, demonstrating the effectiveness of activation clone.

**Alignment-Free Property.** Finally, we evaluate LRC's alignment-free behavior by comparing it to a variant (LRC w/o Alignment Free) that trains additional alignment matrices for attention and FFN outputs. As shown in Figure 3, this variant increases trainable parameter size, prolongs training time, and leads to worse final performance. These results confirm that LRC's projection-based alignment is not only sufficient for effective knowledge transfer but also more efficient and stable.

## 4.4 Model Analysis

To better understand the design choices and behavior of LRC, we conduct a series of in-depth analyses, focusing on two aspects: (1) performance trend during training and (2) impact of training data quality.

**Performance Trend During Training.** We monitor model checkpoints throughout training to examine performance trajectories. Figure 4 shows the variation of MMLU scores, while ARC-C trends are pre-

Table 4: Impact of training data quality.

| Model | LRC-1.5B | | |
|---|---|---|---|
| Teacher | Llama3-3B | | |
| # Tokens | 20B | 10B | 10B |
| Dataset | Mixed-2.0 | Mixed-1.0 | Mixed-1.1 |
| Avg. ↑ | 62.12 | 61.35 | 62.48 |

sented in Appendix E.3. These benchmarks were selected due to their alignment with the overall performance trend observed in Table 1. Results show that LRC achieves competitive performance using just 50% of the training tokens. Moreover, model performance continues to improve steadily with more training, confirming LRC's scalability and efficient learning dynamics.

**Impact of Training Data Quality.** Since LRC requires only a small amount of training data to achieve strong performance, we further examine how training data quality affects performance. Fineweb-Edu [54] provides an educational value score for each sample. To evaluate the impact of higher-quality inputs, we construct a filtered dataset by retaining samples with scores $\geq 4$ and retrain LRC-1.5B using Llama-3.2-3B-Instruct as the teacher. As shown in Table 4, training on this filtered data with just **10B tokens** (Mixed-1.1) surpasses the performance of the **20B-token** setting (Mixed-2.0), both without SFT. This result demonstrates LRC's ability to amplify the benefits of high-quality data, further enhancing its sample efficiency.

**Analysis of Knowledge Transfer in FFNs.** To further investigate why FFNs carry more transferable knowledge than attention, we analyze the distinct roles of these components. Our core hypothesis is that FFNs primarily store factual and world knowledge within their parameters [21, 52], while attention mechanisms are more focused on capturing token-level, contextual relationships, such as syntax and co-reference. Transferring the knowledge embedded in FFNs is therefore especially critical for equipping the student model with foundational understanding.

This view is supported by prior work [21], which identifies FFNs as key-value memories storing knowledge from the pre-training corpus. Specifically, the FFN activation, $\text{act} = \text{SiLU}(zW_{\text{gate}}) \odot zW_{\text{up}}$, measures the similarity between the input $z$ and the knowledge stored in the FFN weights. Our activation clone forces the replication of this similarity distribution, driving the low-rank projection to map the teacher's weights into a similar distribution for the student. Our ablation results in Figure 3 empirically support this view: removing the FFN clone loss (LRC w/o FFN) leads to a significant and persistent drop in performance, while the model recovers more easily from removing the attention clone loss (LRC w/o Attn).

To provide more direct evidence, we conducted a new **neuron-masking experiment** on a factual QA task (e.g., "Who was the first emperor of ancient Rome?" $\rightarrow$ "Augustus"). The procedure is as follows: (1) We input factual questions to the teacher model and identify the top 50 FFN neurons with high activations, termed "important neurons." (2) We masked the **same neuron indices** in the student model's FFNs. (3) As a baseline, we masked 50 random neurons in the student model.

As shown in Table 5, masking the important neurons causes significant performance degradation in both the teacher and student, while random masking has minimal impact. These results confirm that: (1) FFNs encode factual knowledge in specific neurons. (2) LRC effectively transfers this knowledge by aligning the student's activation patterns with the teacher's.

Table 5: Performance of different neuron-masking methods.

| Score Type | Teacher | Student |
|---|---|---|
| Original Score | 0.85 | 0.48 |
| Important Neurons Masked | 0.62 (-27%) | 0.33 (-31%) |
| Random Neurons Masked | 0.85 | 0.49 |

**Compatibility with Other Compression Methods.** LRC is fully compatible with other compression techniques, including structured pruning. To demonstrate this, we applied LLM-Pruner [49] to our LRC-1.5B model, pruning $20\%$ of its parameters. We then conducted a brief 2-hour LoRA [33] fine-tuning to restore performance, resulting in a model we term LRC-Pruned-1.2B.

Table 6: Performance of LRC combined with LLM-Pruner.

| Benchmark | ARC-E | ARC-C | LogiQA | CSQA | PIQA | WinoG | BoolQ | SciQ | MMLU | Avg. ↑ |
|---|---|---|---|---|---|---|---|---|---|---|
| LRC-1.5B | 74.75 | 44.97 | 30.72 | 65.77 | 73.07 | 62.25 | 75.78 | 94.60 | 49.42 | 63.48 |
| **LRC-Pruned-1.2B** | 71.93 | 40.44 | 29.34 | 61.02 | 71.44 | 58.01 | 75.11 | 93.40 | 44.66 | 60.59 |
| MiniCPM-1.2B | 70.16 | 39.68 | 30.88 | 64.29 | 74.65 | 60.77 | 67.58 | 91.50 | 44.23 | 60.42 |

The pruned model outperforms the strong **MiniCPM-1.2B** baseline across most benchmarks, as shown in Table 6. This confirms that LRC can be seamlessly integrated with structured pruning techniques to produce even smaller and more efficient models.

In addition, we study the effects of quantization and the clone loss weighting parameter $\alpha$. Due to space limitations, those results are provided in Appendices E.4 and E.5, respectively.

### 4.5 Efficiency

Finally, we analyze the training efficiency of LRC in terms of memory usage and throughput, focusing on weight sharing strategies and overall training speed.

**Memory-Efficient Weights Sharing.** To further reduce memory overhead and accelerate training, we explore weight sharing across low-rank projection matrices. Specifically, we experiment with tying the projection matrices for input components within both the attention and FFN modules. For attention, we set $\mathbf{W}_q^P = \mathbf{W}_k^P = \mathbf{W}_v^P$, and for FFN, we set $\mathbf{W}_{\text{gate}}^P = \mathbf{W}_{\text{up}}^P$. We train LRC-1.5B on 10B tokens from the Mixed-1.0 dataset using Llama-3.2-3B-Instruct as the teacher and $\alpha = 1.0$. We do not apply SFT to these models.

Table 7 presents the results of four weight-sharing configurations tested in our experiments. The term `All` indicates no weight sharing, while `IO` denotes shared projections for input-only modules. For instance, (`All`, `IO`) signifies no weight sharing in attention but with shared weights in the FFN. The results show that the full-parameter setting (`All`, `All`) delivers the best performance, albeit with the highest memory cost. Notably, sharing projections in the FFN results in a greater performance drop than sharing them in attention. This finding also corroborates the observations from Section 4.3, indicating that FFNs encode richer information and derive greater benefit from dedicated capacity.

Table 7: Performance comparison across different low-rank projection structures of LRC-1.5B.

| (Attn, FFN) | Avg. Score | #Trainable Params | Speedup |
|---|---|---|---|
| (All, All) | **61.22** | 0.93B | 1.00× |
| (IO, All) | 60.81 | 0.67B | 1.07× |
| (All, IO) | 60.25 | 0.80B | 1.05× |
| (IO, IO) | 60.70 | 0.53B | 1.11× |

Table 8: Throughput of training methods on 8 × H800.

| Method | # Tokens/Sec |
|---|---|
| LRC | 84K |
| Sheared Llama (Prune) | 30K |
| Ordinary Training | 146K |
| TinyBERT | 65K |

**Throughput.** Table 8 reports the token-level throughput of LRC training using 8 × H800 GPUs under Zero-2 optimization with `deepspeed`. For reference, we also measure the ordinary pre-training speed of an equivalently sized model using `LlamaFactory` [79]. Despite the overhead of computing the teacher model's hidden states, LRC maintains over 50% of the throughput of standard training.

In contrast, TinyBERT, adapted to the Llama architecture, suffers significantly in throughput, particularly due to its reliance on attention maps as supervision. This requirement prevents the usage of FlashAttention [14], limiting both sequence length and training speed. We also conducted inference throughput tests on vLLM [39], as shown in Appendix E.7. These findings confirm that LRC is not only sample-efficient but also highly practical, offering strong scalability for large-scale training and deployment in real-world settings.

## 5   Conclusions

In this paper, we introduced LRC, a simple yet highly efficient method for distilling SLMs from large teachers. LRC integrates soft pruning and knowledge distillation into a unified framework using trainable low-rank projection matrices. This approach compresses the teacher's weights while simultaneously aligning intermediate activations, with a key focus on the often-overlooked FFN layers. Extensive experiments across diverse downstream tasks demonstrate that LRC models match or outperform state-of-the-art SLMs that were trained on trillions of tokens, despite using up to 1,000× fewer training tokens. These findings position LRC as a promising and resource-efficient paradigm for building compact, high-performing language models.

**Limitations and Broader Impact.** While this study demonstrates the efficiency of LRC under modest training budgets, its performance ceiling under larger-scale training regimes remains unexplored. Additionally, the current implementation retains the same intermediate dimension in the FFN for both teacher and student models, as our primary focus was on distillation efficiency rather than architectural compression. This design choice, however, is not a fundamental constraint of the LRC framework. In fact, LRC is fully compatible with post-hoc compression techniques such as structured pruning [49, 3]. To illustrate this, we applied LLM-Pruner to remove 20% of the parameters from our LRC-1.5B model. The resulting LRC-Pruned-1.2B model still outperforms the strong MiniCPM-1.2B baseline (Table 6), demonstrating the flexibility of LRC in further model compression.

Despite these limitations, LRC offers substantial societal benefits. By drastically reducing the computational cost and data requirements for training high-performing SLMs, it democratizes access to advanced language modeling capabilities. This empowers smaller research groups, academic institutions, and resource-constrained organizations to develop and deploy capable models, fostering broader innovation and inclusivity in the field of AI.

## Acknowledgments and Disclosure of Funding

We would like to thank the anonymous reviewers for their insightful feedback and constructive suggestions, which have significantly improved the quality of this paper. This work was supported by the National Natural Science Foundation of China (NSFC) under Grant Nos. 62125201 and U24B20174.

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

## A  Proof of Lemma 1

**Proof:** Given the Activation Clone conditions:
$$\boldsymbol{h}_{\text{up},i}^{\text{S}} = \boldsymbol{h}_{\text{up},i}^{\text{T}}, \quad \boldsymbol{h}_{\text{gate},i}^{\text{S}} = \boldsymbol{h}_{\text{gate},i}^{\text{T}},$$
the student's FFN output is:
$$\boldsymbol{o}_{\text{ffn},i}^{\text{S}} = \text{SwiGLU}(\boldsymbol{h}_{\text{up},i}^{\text{T}}, \boldsymbol{h}_{\text{gate},i}^{\text{T}}) \boldsymbol{W}_{\text{down},i}^{\text{S}}.$$
Since $\boldsymbol{W}_{\text{down},i}^{\text{S}} = \boldsymbol{W}_{\text{down},i}^{\text{T}} \boldsymbol{W}_{\text{down},i}^{\text{P}}$, we substitute the projection relationship:
$$\boldsymbol{o}_{\text{ffn},i}^{\text{S}} = \text{SwiGLU}(\boldsymbol{h}_{\text{up},i}^{\text{T}}, \boldsymbol{h}_{\text{gate},i}^{\text{T}})(\boldsymbol{W}_{\text{down},i}^{\text{T}} \boldsymbol{W}_{\text{down},i}^{\text{P}})$$
$$= \left( \text{SwiGLU}(\boldsymbol{h}_{\text{up},i}^{\text{T}}, \boldsymbol{h}_{\text{gate},i}^{\text{T}}) \boldsymbol{W}_{\text{down},i}^{\text{T}} \right) \boldsymbol{W}_{\text{down},i}^{\text{P}}.$$
Thus, the student's FFN output exactly matches the teacher's FFN output passed through the same projection matrix. The corresponding MSE loss is:
$$\mathcal{E}(\boldsymbol{o}_{\text{ffn},i}^{\text{S}}, \boldsymbol{o}_{\text{ffn},i}^{\text{T}} \boldsymbol{W}_{\text{down},i}^{\text{P}}) = \mathcal{E}(\boldsymbol{o}_{\text{ffn},i}^{\text{T}} \boldsymbol{W}_{\text{down},i}^{\text{P}}, \boldsymbol{o}_{\text{ffn},i}^{\text{T}} \boldsymbol{W}_{\text{down},i}^{\text{P}}) = 0.$$
This completes the proof. $\qquad\square$

## B  Analytical Scalability of LRC

We believe LRC is scalable and may become even more advantageous as model size increases. The analytical scalability of LRC is supported by the **Johnson-Lindenstrauss (JL) Lemma**, which explains why our low-rank projection becomes more effective for larger models.

- **Formulation:** LRC compresses a teacher's weight matrix $\mathbf{W}^T$ via a projection $\mathbf{W}^S = \mathbf{W}^T \mathbf{W}^P$, where each row of $\mathbf{W}^T$ ($n = d_{\text{ffn}}^T$) is a point in a $d_{\text{model}}^T$-dimensional space. LRC aims to preserve the geometry of these points using Activation Clone.
- **JL Lemma:** The JL Lemma guarantees that this geometry can be preserved if the projected dimension $d^S$ satisfies the condition $d^S \geq O(\log d_{\text{ffn}}^T / \epsilon^2)$, where $\epsilon$ is the error tolerance.
- **Implication for LRC:** As model size increases from 3B to 70B parameters, the actual intermediate dimension of the FFN, $d_{\text{ffn}}^T$, increases moderately (e.g., from 11k to 29k). In contrast, the theoretically required student dimension $d^S$ grows only logarithmically with $d_{\text{ffn}}^T$. This provides a much larger "dimensional budget" for our low-rank projection at larger scales, making it easier to find a high-fidelity projection that preserves the geometric structure of the teacher's weights.

## C  The Pseudo-code of `Forward` **Function**

---
**Algorithm 2:** Transformer Forward Pass (`Forward`)

---
**Input:** Input token sequence $\mathcal{T}$; number of layers $l$; RMSNorm constant $\epsilon$; layer weights $\{\boldsymbol{W}_{\text{q},i}, \boldsymbol{W}_{\text{k},i}, \boldsymbol{W}_{\text{v},i}, \boldsymbol{W}_{\text{o},i}, \boldsymbol{W}_{\text{gate},i}, \boldsymbol{W}_{\text{up},i}, \boldsymbol{W}_{\text{down},i}\}_{i=1}^{l}$; RMSNorm weights $\{\boldsymbol{g}_{\text{attn},i}, \boldsymbol{g}_{\text{ffn},i}\}_{i=1}^{l}, \boldsymbol{g}_{\text{final}}$; embedding weights $\boldsymbol{W}_{\text{emb}}$; LM head weights $\boldsymbol{W}_{\text{lm}}$;

**Output:** Intermediate states dictionary $\boldsymbol{h}$; Attention output dictionary $\boldsymbol{o}_{\text{attn}}$; FFN output dictionary $\boldsymbol{o}_{\text{ffn}}$;

1   $\boldsymbol{h} \leftarrow$ empty dictionary; $\boldsymbol{o}_{\text{attn}} \leftarrow$ empty dictionary; $\boldsymbol{o}_{\text{ffn}} \leftarrow$ empty dictionary;
2   $\boldsymbol{x} \leftarrow \text{Lookup}(\mathcal{T}, \boldsymbol{W}_{\text{emb}})$;
3   **for** $i = 1$ **to** $l$ **do**
     $\triangleright$ `Attention Module`
4     $\boldsymbol{x}_{\text{attn}} \leftarrow \text{RMSNorm}(\boldsymbol{x}, \boldsymbol{g}_{\text{attn},i}, \epsilon)$;
5     $\boldsymbol{h}_{\text{q},i} \leftarrow \boldsymbol{x}_{\text{attn}} \boldsymbol{W}_{\text{q},i}^{\top}; \boldsymbol{h}_{\text{k},i} \leftarrow \boldsymbol{x}_{\text{attn}} \boldsymbol{W}_{\text{k},i}^{\top}; \boldsymbol{h}_{\text{v},i} \leftarrow \boldsymbol{x}_{\text{attn}} \boldsymbol{W}_{\text{v},i}^{\top}$;
6     $\boldsymbol{o}_{\text{attn},i} \leftarrow \text{Attn}(\boldsymbol{h}_{\text{q},i}, \boldsymbol{h}_{\text{k},i}, \boldsymbol{h}_{\text{v},i}) \boldsymbol{W}_{\text{o},i}$;         $\triangleright$ `Store Attention output`
7     $\boldsymbol{x} \leftarrow \boldsymbol{x} + \boldsymbol{o}_{\text{attn},i}$;

     $\triangleright$ `FFN Module`
8     $\boldsymbol{x}_{\text{ffn}} \leftarrow \text{RMSNorm}(\boldsymbol{x}, \boldsymbol{g}_{\text{ffn},i}, \epsilon)$;
9     $\boldsymbol{h}_{\text{gate},i} \leftarrow \boldsymbol{x}_{\text{ffn}} \boldsymbol{W}_{\text{gate},i}^{\top}; \boldsymbol{h}_{\text{up},i} \leftarrow \boldsymbol{x}_{\text{ffn}} \boldsymbol{W}_{\text{up},i}^{\top}$;
10    $\boldsymbol{o}_{\text{ffn},i} \leftarrow \text{SwiGLU}(\boldsymbol{h}_{\text{up},i}, \boldsymbol{h}_{\text{gate},i}) \boldsymbol{W}_{\text{down},i}$;         $\triangleright$ `Store FFN output`
11    $\boldsymbol{x} \leftarrow \boldsymbol{x} + \boldsymbol{o}_{\text{ffn},i}$;

12   **return** $\boldsymbol{h}, \boldsymbol{o}_{\text{attn}}, \boldsymbol{o}_{\text{ffn}}$;

---

Table 9: Model checkpoints used in our experiments.

| Model | Huggingface Model ID |
|---|---|
| InternLM2-1.8B | internlm/internlm2-chat-1_8b |
| Qwen3-1.7B | Qwen/Qwen3-1.7B |
| SmolLM2-1.7B | HuggingFaceTB/SmolLM2-1.7B-Instruct |
| MiniCPM-1.2B | openbmb/MiniCPM-1B-sft-bf16 |
| Gemma3-4B | google/gemma-3-4b-it |
| Qwen3-4B | Qwen/Qwen3-4B |
| Minitron-4B | nvidia/Nemotron-Mini-4B-Instruct |
| Sheared-Llama-2.7B | princeton-nlp/Sheared-LLaMA-2.7B |
| Qwen2.5-7B | Qwen/Qwen2.5-7B-Instruct |
| Qwen2.5-3B | Qwen/Qwen2.5-3B-Instruct |
| Llama3.2-3B | meta-llama/Llama-3.2-3B-Instruct |

# D  Experiment Details

## D.1  Implementation Details of LRC

Using the Llama architecture as our implementation example, we add trainable low-rank projection matrices to the transformer-based structure. For each of the seven weight matrices in the original model corresponding to $q, k, v, o, \text{gate}, \text{up}, \text{down}$, we add a corresponding low-rank projection matrix $\boldsymbol{W}_{\mathrm{m},i}^{\mathrm{p}}$.

During model training, we directly generate the student's weights when performing forward propagation at each layer, and sequentially complete the forward pass for both teacher and student in that layer. We then calculate the clone loss based on the collected intermediate states. This differs slightly from our pseudo-code description but is computationally equivalent.

During initialization, we ensure that only the necessary weights are trained by setting the `requires_grad` attribute to `True` exclusively for the low-rank projection weights and RMSNorm weights of the student.

After training, we use the low-rank projections to transform the teacher's weights into the student's weights. These weights are saved as a new model, and the teacher weights are no longer needed.

## D.2  Checkpoints of Baseline and Teacher Models

In our experiments, most baselines were evaluated directly using `lm-evaluation-harness` on open-source model checkpoints except TinyBert. The detailed configurations of the open-source checkpoints are provided in Table 9. All baseline models utilized the Instruct version, consistent with our choice of the Instruct model for the Teacher.

## D.3  Hyperparameter Settings

We present the hyperparameters used in our experiments in Table 10. Here, "Linear" denotes a scheduler with a warmup stage to the specified learning rate, followed by a linear decay to zero. We used Flash Attention V2 [14] to accelerate training. Notably, the learning rate used for LRC-1.7B is slightly lower than that of other models, as we observed a marginal increase in the number of loss spikes when using $1.0 \times 10^{-4}$. Therefore, the learning rate was reduced in accordance with the decrease in batch size. We adopted a Linear learning rate scheduler, as prior work [8] suggests that this scheduler may be optimal.

## D.4  Model Configurations

Table 11 details the configurations of our LRC models and compares them with their teachers' Llama3.2-3B and Qwen2.5 variants. Key architectural parameters such as layer count, attention heads (Q/KV), hidden/FFN sizes, vocabulary size, and tied embeddings are presented, allowing for direct structural comparison.

Table 10: Training hyperparameters and statistical values in our experiments.

| Model | LRC-1.5B | LRC-1.7B | LRC-4B |
|---|---|---|---|
| Teacher | Llama-3.2-3B-Instruct | Qwen2.5-3B-Instruct | Qwen2.5-7B-Instruct |
| Trained Tokens | 10B | 20B | 18B |
| Pre-train Dataset | Mixed 1.1 | Mixed 1.1 | Mixed 2.0 |
| SFT Dataset | UltraChat | UltraChat | UltraChat |
| Pre-trained Tokens | 10B | 20B | 18B |
| SFT trained Tokens | 0.2B | 0.2B | 0.2B |
| Teacher Hidden Size | 3,072 | 2,048 | 3,584 |
| Student Hidden Size | 1,536 | 1,200 | 2,048 |
| Sequence Length | 2,048 | 2,048 | 2,048 |
| Batch Size (tokens) | 49,152 | 32,768 | 32,768 |
| Clone Loss Weight ($\alpha$) | 0.2 | 0.5 | 0.5 |
| Learning Rate (Pre-train) | $1.0 \times 10^{-4}$ | $6.7 \times 10^{-5}$ | $1.0 \times 10^{-4}$ |
| Learning Rate (SFT) | $1.0 \times 10^{-5}$ | $1.0 \times 10^{-5}$ | $1.0 \times 10^{-5}$ |
| LR Scheduler | Linear | Linear | Linear |
| Warm-up Ratio | 0.005 | 0.005 | 0.005 |
| Optimizer | Adam | Adam | Adam |
| Adam $\beta_1$ | 0.9 | 0.9 | 0.9 |
| Adam $\beta_2$ | 0.999 | 0.999 | 0.999 |
| Temperature for $\mathcal{L}_{\text{KL}}$ | 40 | 40 | 40 |
| RMSNorm $\epsilon$ | $1.0 \times 10^{-5}$ | $1.0 \times 10^{-5}$ | $1.0 \times 10^{-5}$ |
| GPUs | $8 \times$ H800 | $8 \times$ H800 | $8 \times$ H800 |
| Training Time | 34 Hours | 80 Hours | 138 Hours |

Table 11: Model configuration comparison.

| Model | LRC-1.5B | Llama3.2-3B | LRC-1.7B | Qwen2.5-3B | LRC-4B | Qwen2.5-7B |
|---|---|---|---|---|---|---|
| #Layers | 28 | 28 | 36 | 36 | 28 | 28 |
| #Attn Q Heads | 24 | 24 | 16 | 16 | 28 | 28 |
| #Attn KV Heads | 8 | 8 | 2 | 2 | 4 | 4 |
| Head Dim | 128 | 128 | 128 | 128 | 128 | 128 |
| Hidden Size | 1,536 | 3,072 | 1,200 | 2,048 | 2,048 | 3,584 |
| FFN Intermediate Size | 8,192 | 8,192 | 11,008 | 11,008 | 18,944 | 18,944 |
| RMSNorm $\epsilon$ | $1.0 \times 10^{-5}$ | $1.0 \times 10^{-5}$ | $1.0 \times 10^{-6}$ | $1.0 \times 10^{-6}$ | $1.0 \times 10^{-6}$ | $1.0 \times 10^{-6}$ |
| Vocab Size | 128,256 | 128,256 | 151,936 | 151,936 | 152,064 | 152,064 |
| Tie Word Embeddings | True | True | True | True | False | False |

## D.5 Training Dataset Composition

We list all datasets used in our experiments, including Mixed-1.0, Mixed-1.1, Mixed-2.0, along with detailed usage quantities in Table 12. These mixed datasets are based on open-source datasets. The Redpajama data was included to enable fair comparison with Sheared Llama.

All data used in the experiments are open-source, and their corresponding Huggingface data IDs are listed in Table 13.

We randomly sampled 10B high-quality Fineweb-edu data and combined it with the complete OpenHermes dataset to create Mixed-1.0. Building on this, we developed Mixed-1.1 by filtering 20B tokens with an `edu_score` of at least 4, while still utilizing the entire OpenHermes dataset. Since Fineweb-edu primarily targets educational content, we recognized potential limitations in distributional diversity. To address this, we incorporated DCLM, a more diverse dataset containing additional dialogue data. We also integrated Cosmopedia V2, a high-quality synthetic dataset, to further enhance overall data quality. These efforts culminated in the creation of the Mixed-2.0 dataset.

All data were uniformly shuffled. Not all generated data is necessarily used for training. For cost considerations, we sometimes use only a subset of the mixed data.

Table 12: Training dataset composition (# Tokens).

| Training Dataset | Mixed-1.0 | Mixed-1.1-Qwen | Mixed-1.1-Llama | Mixed-2.0 | Redpajama |
|---|---|---|---|---|---|
| Fineweb-Edu | 10B | 20B | 10B | 18B | 0 |
| DCLM | 0 | 0 | 0 | 2B | 0 |
| Cosmopedia V2 | 0 | 0 | 0 | 1B | 0 |
| OpenHermes 2.5 | 450M | 450M | 450M | 450M | 0 |
| Redpajama | 0 | 0 | 0 | 0 | 10B |
| Total | 10.5B | 20.5B | 10.5B | 21.5B | 10B |

Table 13: Open source datasets used in our experiments.

| Dataset | Huggingface Data ID |
|---|---|
| Fineweb-Edu | `HuggingFaceTB/smollm-corpus/fineweb-edu-dedup` |
| Comopedia V2 | `HuggingFaceTB/smollm-corpus/cosmopedia-v2` |
| DCLM | `mlfoundations/dclm-baseline-1.0` |
| OpenHermes-2.5 | `teknium/OpenHermes-2.5` |
| Redpajama | `togethercomputer/RedPajama-Data-1T` |
| UltraChat | `HuggingFaceH4/ultrachat_200k` |

## D.6 Downstream Task and Evaluation Metric Details

We present the evaluation metrics for the tasks used in our evaluation, primarily following the metrics established in Sheared Llama [70]. Evaluation metrics for different downstream tasks and benchmarks are summarized in Table 14.

We selected a diverse suite of datasets spanning four core categories: Scientific and Logical Reasoning (e.g., ARC, LogiQA), Commonsense Understanding (e.g., CSQA, PIQA, WinoGrande), Reading Comprehension (e.g., BoolQ), and World Knowledge (e.g., SciQ, MMLU). This selection, primarily following the established practices in prior work such as Sheared Llama [70] and Minitron [53], is motivated by the need for a comprehensive and multifaceted evaluation of our model's capabilities. Tasks within Scientific and Logical Reasoning directly probe the model's capacity for complex inference, causal understanding, and the application of logical principles, which are crucial for sophisticated problem-solving. Commonsense Understanding benchmarks assess the model's grasp of everyday situations and implicit human knowledge, a vital component for generating natural and coherent interactions. Reading Comprehension tasks evaluate the fundamental ability to extract and synthesize information from text, a cornerstone of language understanding. Finally, World Knowledge datasets measure the breadth and depth of the model's acquired factual information across various domains, reflecting its ability to recall and utilize knowledge effectively. Collectively, these categories provide a holistic view of the model's cognitive strengths and limitations across different facets of intelligence.

While tasks such as mathematical reasoning and code generation are important benchmarks for LLMs, we have deliberately excluded them from our primary evaluation suite for two main reasons. Firstly, for these types of downstream tasks, which often have easily verifiable solutions, there is a possibility that current proprietary LLMs have been pre-trained on extensive, high-quality synthetic datasets specifically curated for these domains. This potential inclusion of specialized synthetic data in their pre-training corpora makes it challenging to draw fair comparisons regarding the inherent capabilities developed through general pre-training, as performance could be heavily skewed by access to such data [74]. Secondly, abilities in mathematics and coding are known to be substantially improvable through post-training alignment techniques, most notably reinforcement learning [25]. As our research primarily focuses on the efficiency and effectiveness of the pre-training phase itself, evaluating tasks whose performance is heavily influenced by subsequent post-training optimization stages falls outside the intended scope of this work. Our evaluation, therefore, centers on tasks that better reflect the foundational knowledge and reasoning abilities acquired directly from the pre-training process on more general textual data.

Table 14: Evaluation metrics for different downstream tasks and benchmarks.

| Downstream Task | Benchmark | Evaluation Metric |
|---|---|---|
| Scientific and Logical Reasoning | ARC-E | Accuracy |
| | ARC-C | Accuracy Norm |
| | LogiQA | Accuracy Norm |
| Commonsense Understanding | CSQA | Accuracy |
| | PIQA | Accuracy |
| | WinoG | Accuracy |
| Reading Comprehension | BoolQ | Accuracy |
| World Knowledge | SciQ | Accuracy |
| | MMLU | Accuracy |
| Safety or Honesty | ToxiGen | Accuracy Norm |
| | TruthfulQA | MC2 |
| Instruction Following | IFeval | Instance-Level Loose Accuracy |

Table 15: Performance of LRC models on general downstream tasks before and after SFT.

| Model | LRC-1.5B | LRC-1.5B-B | LRC-1.7B | LRC-1.7B-B | LRC-4B | LRC-4B-B |
|---|---|---|---|---|---|---|
| ARC-E | 74.75 | 73.40 | 74.62 | 69.49 | 78.37 | 78.75 |
| ARC-C | 44.97 | 42.15 | 44.20 | 42.75 | 52.47 | 52.22 |
| LogiQA | 30.72 | 31.03 | 30.88 | 33.26 | 34.10 | 34.87 |
| CSQA | 65.77 | 64.46 | 70.19 | 70.27 | 79.28 | 78.30 |
| PIQA | 73.07 | 71.60 | 73.07 | 71.38 | 76.82 | 76.61 |
| WinoG | 62.25 | 61.88 | 63.30 | 63.85 | 67.72 | 67.80 |
| BoolQ | 75.78 | 73.27 | 79.82 | 75.78 | 84.50 | 84.95 |
| SciQ | 94.60 | 94.40 | 93.80 | 89.00 | 95.00 | 94.30 |
| MMLU | 49.42 | 50.09 | 54.93 | 55.13 | 64.41 | 64.58 |
| Avg. | 63.48 | 62.48 | 64.98 | 63.43 | 70.30 | 70.26 |

## D.7 Implementation Details of TinyBERT

Since TinyBERT requires using attention score maps as training supervision labels, we cannot use Flash Attention [14], so we had to reduce the max sequence length to 512 to decrease memory usage and improve training efficiency. All other experimental settings are fully aligned, including the student's total parameter count, number of training tokens, learning rate, and other hyperparameters.

## D.8 Details of Experiments about Activation Clone

When testing the module-level impact of different clone losses on LM loss convergence in Activation Clone, we used `IO Attn` rather than the better-performing `All Attn`. The definition of `IO Attn` can be found in Section 4.5. This choice was necessary because our experiments revealed that training with `All Attn` becomes highly unstable without the constraint provided by clone loss. Therefore, we were limited to using `IO Attn` for our analysis.

# E  Additional Experiments

## E.1  Impacts of SFT on Model Performance

Modern LLMs typically undergo a two-phase training process: pre-training followed by post-training [22, 74], where post-training focuses on instruction following, alignment with human preferences, and safety. We compared the performance changes of the LRC model after SFT. To this end, we evaluate the student models on three widely used safety/honesty and instruction-following benchmarks: ToxiGen [28], TruthfulQA [43], and IFeval [81]. The specific metrics are also listed in Table 14.

Table 16: Performance of LRC models on safety and instruction-following tasks before and after SFT.

| Model | LRC-1.5B | LRC-1.5B-B | LRC-1.7B | LRC-1.7B-B | LRC-4B | LRC-4B-B |
|---|---|---|---|---|---|---|
| **ToxiGen** | 43.19 | 43.19 | 43.30 | 43.30 | 43.72 | 43.83 |
| **IFeval** | 23.74 | 24.58 | 39.69 | 36.69 | 13.67 | 36.09 |
| **TruthfulQA** | 46.98 | 47.97 | 47.95 | 53.17 | 50.71 | 55.89 |

Table 17: 5-shot model performance on various benchmarks.

| Benchmark | WinoGrande | ARC-C | BoolQ | MMLU | Avg. ↑ |
|---|---|---|---|---|---|
| **Gemma3-4B** | 69.06 | 60.49 | 84.77 | 58.33 | 68.16 |
| **Minitron-4B** | 73.95 | 53.58 | 82.39 | 57.86 | 66.95 |
| **Qwen3-4B** | 66.85 | 61.18 | 85.27 | 70.04 | 70.84 |
| **LRC-4B** | 69.93 | 58.36 | 85.69 | 65.10 | 69.77 |
| **InternLM2-1.8B** | 65.27 | 44.03 | 78.59 | 45.99 | 58.47 |
| **LRC-1.7B** | 63.38 | 48.98 | 81.74 | 54.83 | 62.23 |
| **Qwen3-1.7B** | 60.62 | 52.22 | 80.61 | 60.15 | 63.40 |
| **SmolLM2-1.7B** | 69.14 | 51.88 | 75.11 | 49.32 | 61.36 |
| **LRC-1.5B** | 60.77 | 47.95 | 79.24 | 50.68 | 59.66 |
| **MinCPM-1.2B** | 64.80 | 44.71 | 76.45 | 48.68 | 58.66 |

The observed improvement in general downstream tasks post-SFT, shown in Table 15, contrasts with the limited gains in safety benchmarks and a decline in instruction-following for LRC models, which is shown in Table 16. This divergence likely stems from the composition and inherent limitations of the SFT dataset (UltraChat). While SFT enhances knowledge and common task execution, its efficacy for nuanced capabilities like complex instruction adherence and robust safety alignment is more constrained. The SFT data may lack sufficient diversity or targeted examples for these specialized domains. For instance, IFEval's intricate instructions might not be well-represented, potentially leading the model to prioritize fluency or common response patterns learned during SFT over the precise execution of novel, complex directives.

Similarly, without a substantial corpus of safety-focused demonstrations and negative examples, significant improvements in benchmarks like ToxiGen are unlikely. Although TruthfulQA shows some gains, possibly from increased factuality within the SFT data, the overall pattern suggests that achieving strong instruction-following and safety often necessitates more targeted data or advanced alignment techniques, such as Reinforcement Learning from Human Feedback (RLHF), which are specifically designed to instill these fine-grained behaviors more effectively than standard SFT.

## E.2 Few-Shot Results and Analyses

Following previous works [70, 53], we additionally evaluated the performance of LRC on few-shot tasks, with results presented in Table 17. The findings indicate that LRC models exhibit more modest performance gains in few-shot scenarios compared to baseline models. This is particularly notable given LRC's strong zero-shot capabilities, where it often surpasses competitors like Qwen.

Several factors may contribute to this observation. Firstly, as initially posited, models such as Qwen3 and SmolLM2 might benefit from post-training strategies involving increased training data length or the specific collection and construction of data for long-context scenarios. Such data could implicitly bolster their proficiency in in-context learning. Secondly, the superior few-shot adaptability of baseline models could be attributed to more extensive SFT or instruction tuning phases, which are specifically designed to enhance a model's capacity to learn from a small number of examples. Consequently, while LRC's pre-training fosters robust zero-shot generalization, its architecture or training objectives may not be as readily optimized for the distinct skill of rapid adaptation from few-shot demonstrations. We plan to investigate these potential factors in future work, including a closer examination of baseline training methodologies and exploring targeted fine-tuning for LRC to improve its few-shot performance.

### E.3 Performance Trend of ARC-C during Training

As shown in Figure 5, the trend exhibited by the ARC-C is generally consistent with that of MMLU. These two results together demonstrate the effectiveness of the LRC method and showcase its scalability.

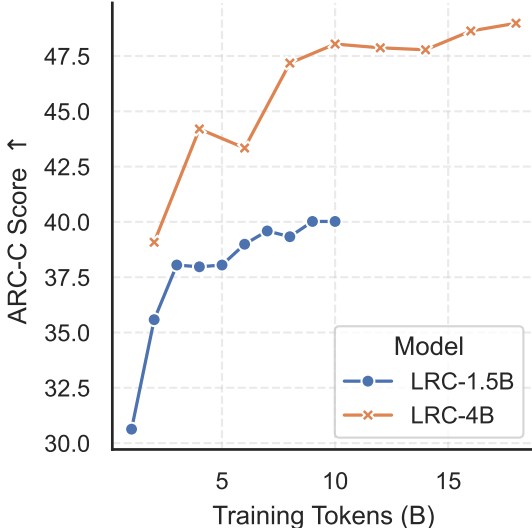

Figure 5: The trend of ARC-C scores with increasing training tokens.

### E.4 Impact of Quantization

We further investigate whether models trained with LRC can be combined with quantization techniques to further reduce memory requirements. We utilize `bitsandbytes` [17] to perform 8-bit quantization on our model, and the experimental results are shown in Table 18. The experimental results indicate that the performance loss of our LRC-trained model remains within acceptable limits, which, to some extent demonstrates that the numerical range of our model is within normal parameters and does not significantly impact the quantization method.

Table 18: Impact of quantization using `bitsandbytes` [17] on model performance.

| Model | SmolLM2-1.7B | | LRC-1.5B | | LRC-1.7B | |
|---|---|---|---|---|---|---|
| Quantization | INT8 | BF16 | INT8 | BF16 | INT8 | BF16 |
| ARC-E | 70.24 | 69.11 | 74.79 | 74.75 | 73.82 | 74.62 |
| ARC-C | 43.26 | 43.52 | 44.71 | 44.97 | 43.86 | 44.20 |
| LogiQA | 26.88 | 28.88 | 30.11 | 30.72 | 31.03 | 30.88 |
| CSQA | 50.12 | 51.19 | 65.27 | 65.77 | 70.68 | 70.19 |
| PIQA | 75.41 | 76.01 | 73.01 | 73.07 | 72.58 | 73.07 |
| WinoG | 67.64 | 68.98 | 61.88 | 62.25 | 62.90 | 63.30 |
| BoolQ | 68.99 | 68.47 | 76.15 | 75.78 | 79.69 | 79.82 |
| SciQ | 89.50 | 89.80 | 94.20 | 94.60 | 93.50 | 93.80 |
| MMLU | 47.38 | 48.50 | 49.13 | 49.42 | 53.88 | 54.93 |
| Avg. ↑ | 59.94 | 60.50 | 63.25 | 63.48 | 64.66 | 64.98 |

### E.5 Choice of $\alpha$

The hyperparameter $\alpha$ controls the relative strength of the activation clone loss $\mathcal{L}_{\text{clone}}$. We experiment with different values of $\alpha$ and report performance in Figure 6. The results exhibit an "n"-shaped trend: When $\alpha$ is small, the clone loss fails to adequately guide the student to imitate the teacher's behavior; When $\alpha$ is extremely large, training becomes unstable due to large gradient norms.

One possible explanation is that the mismatch in parameter capacity between teacher and student makes over-enforcing behavioral similarity counterproductive. We leave a deeper exploration of this phenomenon to future work.

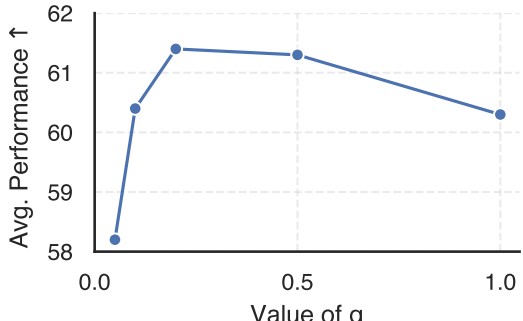

Figure 6: The impact of $\alpha$ on model average performance.

## E.6  Clone Loss as Anchor

We investigated the impact of removing clone loss from certain layers on the convergence rate of LM loss, aiming to verify whether each layer's clone loss accelerates training. We tested removing $50\%, 25\%, 12.5\%$ of the layers and observed the convergence behavior of the LM loss, with experimental results shown in Figure 7. This experiment demonstrates that the clone loss at each layer is important for LM loss convergence. These clone losses applied to each module in each layer can be viewed as anchor points, constraining the behavior of each student module to remain similar to its teacher counterpart.

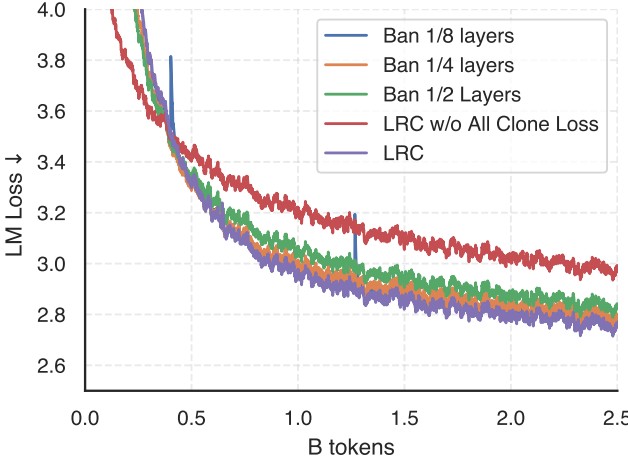

Figure 7: The impact of removing the clone loss from certain layers on the convergence of LM loss.

## E.7  Inference Speed

We tested the inference speed on vLLM [39], with results shown in Table 19. From the perspective of inference speed, our model achieves a good level of performance while also maintaining satisfactory inference speeds.

## E.8  Result Stability with Different Random Seeds

We agree that evaluating variability is crucial for demonstrating the robustness and reliability of our method. While the high computational cost of pre-training limits our ability to perform extensive multi-run experiments across all settings, we conducted a second run of our key LRC-1.5B experiment using a different random seed (42) to assess result stability. The performance across both runs is presented in Table 20.

Table 19: Inference throughput of vLLM.

| Throughput | # Input Tokens/Sec | # Output Tokens/Sec |
|---|---|---|
| LRC-1.5B | 68,223 | 15,574 |
| Qwen3-1.7B | 71,176 | 15,404 |
| SmolLM2-1.7B | 48,285 | 10,491 |
| MiniCPM-1.2B | 58,136 | 13,191 |

Table 20: Stability of LRC under different random seeds.

| Benchmark | ARC-E | ARC-C | LogiQA | CSQA | PIQA | WinoG | BoolQ | SciQ | MMLU | Avg. ↑ |
|---|---|---|---|---|---|---|---|---|---|---|
| LRC-1.5B (Seed 218) | 74.75 | 44.97 | 30.72 | 65.77 | 73.07 | 62.25 | 75.78 | 94.60 | 49.42 | 63.48 |
| **LRC-1.5B (Seed 42)** | 76.81 | 42.75 | 30.72 | 63.64 | 72.31 | 62.43 | 77.86 | 94.10 | 49.05 | 63.30 |
| MiniCPM-1.2B | 70.16 | 39.68 | 30.88 | 64.29 | 74.65 | 60.77 | 67.58 | 91.50 | 44.23 | 60.42 |

The performance variance between the two runs is minimal (<0.2). This result increases our confidence in the robustness of LRC. Due to time constraints, we were only able to conduct one additional run. We plan to conduct further runs in the future to provide more comprehensive mean and variance statistics.

## E.9 Structural Properties of Low-Rank Projection Matrices

Our hypothesis is that for general-purpose distillation, the low-rank projection in LRC is primarily **structure-preserving**, aiming to retain the rich and diverse capabilities of the teacher model rather than selectively pruning information. This is essential for cloning a broad range of knowledge encoded in the teacher's parameters. To investigate this, we conducted two complementary analyses:

**SVD Analysis.** We tracked the singular values of a representative projection matrix, $\mathbf{W}_{up}^{P}$, during training. As shown in Table 21, the singular values increase across training, suggesting that the projection matrix retains a high-rank structure and does not collapse into a small subspace. This indicates that the projection actively utilizes the full dimensionality of the teacher model, supporting the goal of comprehensive knowledge transfer.

Table 21: Evolution of singular values during training.

| Singular Rank Percentage | 0% | 10% | 50% | 90% | 100% |
|---|---|---|---|---|---|
| Train. 10% | 6.14 | 2.48 | 1.72 | 0.79 | 0.37 |
| Train. 50% | 6.69 | 3.62 | 2.74 | 1.61 | 0.42 |
| Train. 100% | 6.70 | 3.85 | 2.88 | 1.67 | 0.38 |

Table 22: Structural Similarity of FFN Weights.

| Exp. Type | MSE (Teacher vs. Student) | MSE (Teacher vs. Random) |
|---|---|---|
| $\mathbf{W}_{up}$ | 0.000576 | 0.001026 |
| $\mathbf{W}_{gate}$ | 0.000635 | 0.001202 |
| $\mathbf{W}_{down}$ | 0.000559 | 0.001124 |

**Structural Similarity Analysis.** We evaluated whether LRC preserves the internal weight geometry of the teacher by comparing similarity matrices $\mathbf{Sim} = \mathbf{W}\mathbf{W}^{T}$ of the FFN weights. Specifically, we computed the Mean Squared Error (MSE) between the teacher and student similarity matrices and compared this to a baseline with randomly initialized weights. As shown in Table 22, LRC significantly reduces the structural discrepancy compared to the random baseline, demonstrating that the student retains the internal structural patterns of the teacher.

While we believe selective pruning is less likely for general distillation, we see this as an exciting direction for future work, where LRC could be extended to specialize in specific tasks or domains.

