# OpenReview forum: "A Token is Worth over 1,000 Tokens: Efficient Knowledge Distillation through Low-Rank Clone"
_NeurIPS.cc/2025/Conference — NeurIPS 2025 spotlight_

### Official Review · Reviewer_Vtru · 2025-06-24

**Clarity:** 3
**Significance:** 3
**Originality:** 3
**Rating:** 5
**Confidence:** 3

**Summary:**

This paper presents an efficient knowledge distillation framework to train a high-performing small language model from a large one. The proposed approach combines two main techniques: layer weight distillation, which introduces a set of low-rank projection matrices to softly compress and distill teacher weights, and activation cloning, which aligns the student’s intermediate activations—particularly the feed-forward network (FFN) outputs—with those of the teacher. Extensive experiments on various open-source teacher models demonstrate both the effectiveness and computational efficiency of the proposed method.

**Questions:**

Please see weakness.

**Ethical Concerns:**

["NO or VERY MINOR ethics concerns only"]

**Final Justification:**

My initial concers have been solved. I maintain my previous rating.  The authors are encouraged to  hightlight that activation Clone loss is the primary driver of knowledge transfer in LRC.

**Limitations:**

Yes

**Quality:**

3

**Strengths And Weaknesses:**

Strengths:
1. This paper is clearly presented and well motivated, addressing the small language model's training.
2. Experiments are done thouroughly to assess the performance of proposed method across different settings.
3. Code is well organized and included in submission, which enhances reproducibility.

Weakness:
1. Some related works are missing in discussion. In particular
[1]LIMA: Less Is More for Alignment
[2]DA-KD: Difficulty-Aware Knowledge Distillation for Efficient Large Language Models
including discussion against these works would strenghthen the techinical contribution.

2. The authors mention "Minimal Information Loss", yet xcept the final performance, any metric to support this claim? For example, have the authors tried feeding the input student's output to teacher to evaluate how well it retains the teacher's internal representations?

3. In eq(4),  includes a KL loss, a vanilla knowledge distillation, It would be useful to report ablation results without this term to understand the constribution of the proposed loss alone.

---

> ### Author Rebuttal · Authors · 2025-07-31
>
> Thank you for your supportive and constructive review. We are glad that you found our paper “clearly presented and well motivated” and appreciated the “thorough” experimental validation. Your thoughtful feedback is greatly appreciated and has been instrumental in helping us further refine and strengthen our work.
>
> Below, we respond to the concerns and questions you raised.
>
> ---
>
> > **W1: Some related works are missing in discussion.**
>
> Thank you for pointing out these important related works. We agree that discussing them will help clarify the scope and novelty of LRC’s contributions. We will incorporate a detailed discussion into the revised manuscript. Below, we outline how LRC relates to these efforts:
>
> - Regarding LIMA [1]: LIMA compellingly demonstrates that model alignment may depend more on a small, high-quality dataset rather than sheer data volume. LIMA's contribution lies in data selection for the alignment phase, whereas our LRC focuses on knowledge distillation during the pre-training phase. Therefore, LIMA's methodology is orthogonal to LRC. We believe that combining an LRC-distilled model with LIMA's high-quality data strategy during alignment could lead to even greater efficiency and performance.
> - Regarding DA-KD [2]: DA-KD introduces a "Difficulty-Aware Knowledge Distillation" framework. Similar to LIMA, DA-KD's innovation is at the data level. In contrast, LRC's innovation is at the model level--efficiently extracting knowledge directly from the teacher's weights and activations. Consequently, DA-KD's data selection framework can be viewed as a complementary technique. Training an LRC model on data filtered by DA-KD could potentially further enhance LRC's distillation efficiency.
>
>
> > **W2: Any metric to support the "Minimal Information Loss" claim?**
>
> We admit that "Minimal Information Loss" might be too strong a term. Our intention was to convey that our "soft pruning" via low-rank projection preserves more information from the teacher's weights compared to "hard pruning," which discards weights entirely before training. We will revise the wording to be more precise, such as "Reduced Information Loss."
>
> To quantitatively support the claim that our method effectively transfers knowledge with low loss, we conducted a **new neuron-masking experiment** on a factual QA task (e.g., "Who was the first emperor of ancient Rome?" → "Augustus"):
>
> 1. We input factual questions to the teacher model and identified the top 50 FFN neurons with the highest activations, termed "important neurons."
> 2. We masked the **same neuron indices** in the student model's FFNs.
> 3. As a baseline, we masked 50 random neurons in the student model.
>
> ***Table R1:** Performance of different neuron-masking methods (teacher: Llama-3.2-3B, student: LRC-1.5B).*
>
> |Score Type|Teacher|Student|
> |:---|:---|:---|
> |Original Score|0.85|0.48|
> |**Important Neurons Masked**|**0.62 (-27%)**|**0.33 (-31%)**|
> |Random Neurons Masked|0.85|0.49|
>
> As shown in Table R1, masking the important neurons causes significant performance degradation in both teacher and student, while random masking has minimal impact. These results confirm that:
>
> - FFNs encode localized factual knowledge in specific neurons;
> - **LRC effectively transfers this knowledge by aligning the student's activation patterns with the teacher's.**
>
>
> We also measured the **Structural Similarity** if LRC preserves the internal weight geometry of the teacher by comparing similarity matrices $\text{Sim}=\mathbf{W}\mathbf{W}^\top$ of the FFN up, gate, and down projections. Specifically, we computed the MSE between the teacher and student similarity matrices, and compared this to a baseline with randomly initialized weights.
>
> As shown in Table R2 below, LRC significantly reduces the structural discrepancy compared to the random baseline, demonstrating that the student retains internal structural patterns of the teacher.
>
> ***Table R2:** Structural Similarity of FFN Weights.*
> |Exp. Type|MSE (Teacher vs. Student)|MSE (Teacher vs. Random)|
> |:--|:--|:--|
> |$\mathbf{W}_{\text{up}}$|**0.000576**|0.001026|
> |$\mathbf{W}_{\text{gate}}$|**0.000635**|0.001202|
> |$\mathbf{W}_{\text{down}}$|**0.000559**|0.001124|
>
>
> > **W3: It would be useful to report ablation results without the KL loss term.**
>
> As described in Equation 4 of our manuscript, our total training objective combines three components: (i) standard KL divergence on output logits $\mathcal{L}\_\text{KL}$, (ii) next-token prediction loss $\mathcal{L}\_\text{LM}$, and (iii) Activation Clone loss $\mathcal{L}\_\text{clone}$.
>
> To isolate the impact of the KL term, we ran a new ablation experiment by setting the weight of $\mathcal{L}\_\text{KL}$ to zero. Table R3 below reports the LM training loss at different steps:
>
> ***Table R3:** Ablation study of KL loss.*
>
> | Training Steps | LRC (w/o KL Loss) | LRC (Full) |
> | :--- | :--- | :--- |
> | 10K | 3.296 | 3.303 |
> | 20K | 2.952 | 2.949 |
> | 30K | 2.853 | 2.849 |
> | 40K | 2.784 | 2.786 |
> | 50K | 2.735 | 2.745 |
>
> As shown in Table R3, removing the KL loss results in only marginal differences in the training loss across all stages. This indicates that the **Activation Clone loss is the primary driver of knowledge transfer in LRC**, effectively guiding the student to replicate the teacher’s internal representations. This confirms that the effectiveness of activation cloning mechanisms.
>
> ---
>
> Thank you for taking the time to provide us with your thoughtful and constructive feedback. We hope that our responses address your concerns, and the new experiments will be included in the revised manuscript. We welcome any further questions you may have.

---

> > ### Comment · Reviewer_Vtru · 2025-08-04
> >
> > Thanks for the response. My initial concerns are solved.

---

> > > ### Author Response · Authors · 2025-08-05
> > >
> > > Thank you for your follow-up and for taking the time to review our work. We are glad to hear that your initial concerns have been resolved. If you have any additional questions or suggestions, please feel free to reach out. We are happy to continue the discussion and provide further clarifications.
> > >
> > > Thanks again for your thoughtful feedback and support!

---

### Official Review · Reviewer_4MPA · 2025-06-30

**Clarity:** 3
**Significance:** 3
**Originality:** 4
**Rating:** 4
**Confidence:** 4

**Summary:**

This work introduces a novel LLM soft shrinking method that utilizes trainable projections from the teacher to the student dimension to create student weights. Moreover, they extend it with knowledge distillation and show that, as we already have shrinking projections, we can conveniently utilize them to perform hidden state matching in a more efficient way. Finally, they show that this algorithm is much more token-efficient than standard pretraining from scratch and more time-efficient than TinyBert (distillation with no pruning).

**Questions:**

1. How does this method compare to iso-FLOP state-of-the-art distillation initialized with pruning (like Minitron/DistillGPT)? (see Weaknesses)
2. Does this method keep the size of the Feed Forward constant from the teacher? If it does, this should be mentioned in the Limitations section. (see Limitations)

**Ethical Concerns:**

["NO or VERY MINOR ethics concerns only"]

**Final Justification:**

This paper introduces a novel, promising soft-shrinking technique that appears to be a good alternative to the dominant shrinking + distillation recipe. However, the initial submission did not contain a proper validation confirming the validity of the method, comparing their results to distillation only (while the low-rank clone model was also distilled). Although the authors submitted this during the rebuttal period, this strongly changes the conclusions of the work, as a low-rank clone is only superior for long horizons, which are not always practical for shrinking. Moreover, the paper did not mention that this method does not decrease the width of feed foward and attention head size, what also may become problematic for hierarhical shrinking, which is the most practical application. Therefore, while in my view this work deserves recogniction, the way it was written (and especially validations) raise a lot of concerns.

**Limitations:**

1. If I understood it correctly, this method keeps the dimensionality of the hidden size in the Feed Forward network and the dimensionality of attention from the teacher, constraining the shape of the student significantly. I think this is an important limitation, and it was not mentioned in the Limitations section, and was only indirectly deducible from the assumptions in the construction of these projections.

**Paper Formatting Concerns:**

I did not notice any.

**Quality:**

2

**Strengths And Weaknesses:**

__Strengths:__
1. The main idea of this work, that is, using trainable projections for LLM shrinking, is novel and seems very promising. These projections introduce token-independent overhead (which I suppose is insignificant with a sufficiently large batch size) and can be multiplied out after training, making this inference efficient. Since teacher weights are frozen, this potentially allows the student to not forget information from the teacher so easily at the beginning, in comparison to initialization with pruning, allowing the student to benefit from it longer. The idea sounds simple and intuitive, yet it is not present im literature according to my knowledge, strengthening the potential impact.
3. This method can be used alongside distillation and makes hidden state matching more efficient, as these projections can be reused to align activations.
4. This work tries to solve an important problem of LLM inference-optimality with a fresh idea that is potentially impactful and does not seem incremental.

__Weaknesses:__
1. Lack of efficiency comparison (e.g., iso-FLOP) to state-of-the-art distillation of the same base teacher model (with and without hidden state matching) initialised with pruning  (like Minitron/DistillGPT) makes this work really hard to judge. While the core idea behind this method seems very promising, it is rather a replacement for pruning, since both can be used alongside distillation. The only efficiency comparison (with real-time) was made with TinyBert, which in itself has very high overhead and does not use pruning at all.
2. While low-rank projections seem like an important contribution and softer replacement for shrinking/pruning, activation clone is presented as a novel contribution, but is rather just a use of low-rank projections with modern distillation, while alignment-free property is a rather natural consequence of using both things at once.
3. The title only multiplies issues with evaluations present in this work, suggesting __1000x__ token efficiency, which immediately raises concerns. Naturally, pretraining from scratch is much less efficient than any shrinking method, especially if the model is radically overtrained with more than 20000 tokens per parameter. I suppose that, if Qwen-1.7B baseline would be trained for another 30T tokens, the performance would not increase significantly, and the title could be about __2000x__ token efficiency, which makes this comparison lacking substance.

---

> ### Author Rebuttal · Authors · 2025-07-31
>
> Thank you for your thoughtful and encouraging feedback.
> We are especially grateful for your recognition of our method as *"a fresh idea that is potentially impactful and does not seem incremental"* and for describing our core concept as *"excellent"* in term of originality. Your insightful questions have helped us clarify and strengthen our paper's contributions. Your kind words are deeply appreciated.
>
> Moreover, your insightful comments and questions have been instrumental in helping us clarify and strengthen the presentation of our contributions. Please find our detailed responses below.
>
> ---
>
> > **W1, Q1: Lack of efficiency comparison to state-of-the-art distillation of the same base teacher model initialized with pruning (like Minitron/DistillGPT).**
>
> Following your recommendation, we conducted a new set of experiments comparing LRC-1.5B with Minitron [1], using an identical setup for fairness. Both methods distilled from the same teacher, Llama-3.2-3B-Instruct, on the same hardware (8× NVIDIA H800 GPUs) and training data.
> - For Minitron, we followed the official pipeline: first applying width pruning to obtain Minitron-1.5B, then performing distillation.
> - For LRC, we applied our proposed low-rank projection and activation cloning approach without any pre-pruning.
>
> We report training dynamics in terms of real-time loss (Table R1) and evaluate zero-shot performance on standard benchmarks (Table R2).
>
> ***Table R1:** Real-time training loss comparison.*
>
> |Training Time|1 Hour|3 Hours|5 Hours|10 Hours|20 Hours|30 Hours|
> |:--|:--|:--|:--|:--|:--|:--|
> |Minitron-1.5B|2.96|2.82|2.75|2.70|2.63|2.62|
> |LRC-1.5B|4.02|2.97|2.79|**2.58**|**2.49**|**2.46**|
>
> ***Table R2:** Zero-shot performance comparison between LRC-1.5B and Minitron-1.5B (First row taken from Table 1 in paper).*
>
> |Benchmark|ARC-E|ARC-C|LogiQA|CSQA|PIQA|WinoG|BoolQ|SciQ|MMLU|**Avg.↑**|
> |:--|:--|:--|:--|:--|:--|:--|:--|:--|:--|:--|
> |LRC-1.5B     |74.75|44.97|30.72|65.77|73.07|62.25|75.78|94.60|49.42|**63.48**|
> |Minitron-1.5B|66.88|34.81|29.80|57.41|70.57|57.41|70.24|93.10|41.60|**58.20**|
>
> As shown in Table R1, while Minitron benefits from weight inheritance and shows lower initial loss, LRC quickly surpasses it, achieving lower loss in under 5 hours and maintaining faster convergence throughout. This reflects the training efficiency of LRC despite not inheriting weights directly.
>
> Moreover, as depicted in Table R2, LRC consistently outperforms Minitron across all benchmarks, with especially large gains on knowledge-intensive tasks like MMLU and CSQA.
>
> These results support our key hypothesis: by combining soft pruning (via low-rank projection) with activation cloning, LRC preserves and transfers rich teacher knowledge more effectively than conventional pruning-based distillation pipelines, while maintaining superior training efficiency.
>
>
> > **W2: Activation clone is a use of low-rank projections with modern distillation and alignment-free property is a rather natural consequence.**
>
> Thank you for highlighting the value of our low-rank projection design. While we agree that the alignment-free property naturally follows from our unified formulation, we argue that both activation cloning and alignment-free learning represent important and non-trivial contributions in their own right.
>
> The novelty of activation clone lies not just in its contribution with low-rank projection, but in demonstrating the critical importance of aligning the FFN activations. As shwon in our ablation study (`LRC w/o FFN` in Figure 3), removing the FFN activation loss leads to a consistent and significant drop in performance, indicating that FFNs are central to knowledge retention and transfer.
>
> To further validate this, we conducted a **new neuron-masking experiment** on a factual QA task (e.g., "Who was the first emperor of ancient Rome?" → "Augustus"):
>
> 1. We input factual questions to the teacher model and identified the top 50 FFN neurons with the highest activations, termed "important neurons."
> 2. We masked the **same neuron indices** in the student model's FFNs.
> 3. As a baseline, we masked 50 random neurons in the student model.
>
> ***Table R3:** Performance of different neuron-masking methods (teacher: Llama-3.2-3B, student: LRC-1.5B).*
>
> |Score Type|Teacher|Student|
> |:---|:---|:---|
> |Original Score|0.85|0.48|
> |**Important Neurons Masked**|**0.62 (-27%)**|**0.33 (-31%)**|
> |Random Neurons Masked|0.85|0.49|
>
> As shown in Table R3, masking the important neurons causes significant performance degradation in both teacher and student, while random masking has minimal impact. These results confirm that:
>
> - FFNs encode localized factual knowledge in specific neurons;
> - **LRC effectively transfers this knowledge by aligning the student's activation patterns with the teacher's.**
>
> As for the alignment-free design, while it may seem like a natural consequence, it brings a practical advantage often overlooked. Traditional distillation methods require learning separate alignment matrices, which add trainable parameters, increase optimization complexity, and can result in unstable convergence. In contrast, our design aligns teacher and student subspaces intrinsically through projection, without the need for additional tuning.
> As shown in the ablation study (`LRC w/o Alignment Free` in Figure 3), removing this simplicity harms both performance and stability, underscoring its importance as a key design decision, not just a byproduct.
>
> We hope this clarifies the conceptual and empirical value of activation cloning and alignment-free learning within LRC.
>
>
>
> > **W3: The title only multiplies issues with evaluations present in this work, suggesting 1000x token efficiency.**
>
> Thank you for raising this important point. We agree that the phrasing in the title could be misinterpreted, especially in light of the complex tradeoffs involved in efficiency evaluation. While our experiments do demonstrate that LRC achieves comparable or better performance than models trained on over 1000× more tokens, we acknowledge that this claim--when presented in the title--can appear overstated or overly simplified.
>
> We will revise the title to more precisely reflect our key contribution: a general and efficient distillation framework that significantly reduces training cost while maintaining strong performance.
>
>
>
> > **Q2, L1: This method keeps the size of the Feed Forward constant from the teacher.**
>
> You are correct that in our current formulation, LRC retains the same FFN intermediate dimension as the teacher model. We appreciate you pointing this out, and we will explicitly mention this assumption in the revision of the paper.
>
> However, we would like to emphasize that this design choice is not a fundamental limitation of the LRC framework. Rather, it reflects our focus on distillation efficiency rather than architectural compression in the current study. In fact, LRC is fully compatible with post-hoc model compression techniques. To illustrate this, we conducted a new experiment applying LLM-Pruner [2], a structured pruning method, to our LRC-1.5B model. We pruned 20% of the parameters, followed by a brief 2-hour LoRA fine-tuning phase, resulting in LRC-Pruned-1.2B. The results are shown in Table R4.
>
> ***Table R4:** New results with LLM-Pruner.*
>
> |Benchmark|ARC-E|ARC-C|LogiQA|CSQA|PIQA|WinoG|BoolQ|SciQ|MMLU|Avg.↑|
> |:---|:---|:---|:---|:---|:---|:---|:---|:---|:---|:---|
> |LRC-1.5B|74.75|44.97|30.72|65.77|73.07|62.25|75.78|94.60|49.42|63.48|
> |**LRC-Pruned-1.2B**|71.93|40.44|29.34|61.02|71.44|58.01|75.11|93.40|44.66|60.59|
> |MiniCPM-1.2B|70.16|39.68|30.88|64.29|74.65|60.77|67.58|91.50|44.23|60.42|
>
> As shown above, LRC-Pruned-1.2B outperforms MiniCPM-1.2B despite being of similar size, validating that LRC's distilled representations remain robust even under structural pruning. This result also suggests that our FFN dimension can be adjusted post-training without degrading model quality, effectively addressing your concern.
>
> For future work, we plan to explore more integrated compression strategies within the LRC framework. For instance, introducing an additional low-rank projection in the FFN layers could allow us to reduce the intermediate dimension during distillation, enabling even more compact and efficient student models.
>
> ---
>
> Thank you again for your valuable and thorough feedback. We hope the newly added analyses and experiments have addressed your concerns. We will include these additional results in the revised manuscript. Please let us know if you have any other questions or concerns.
>
> [1] Compact Language Models via Pruning and Knowledge Distillation.
>
> [2] LLM-Pruner: On the Structural Pruning of Large Language Models.

---

### Official Review · Reviewer_sUgd · 2025-07-02

**Clarity:** 3
**Significance:** 3
**Originality:** 2
**Rating:** 5
**Confidence:** 3

**Summary:**

The paper proposes **Low-Rank Clone (LRC)**, a two-stage framework that lets a small language model inherit knowledge from a larger teacher:

1. Low-Rank Projection: For every transformer layer, the teacher’s large weight matrices are factorised through learnable low-rank projectors; the projected (compressed) matrices are used directly as the student’s parameters, so the student is generated on-the-fly rather than trained from scratch. Only the projector weights and RMSNorm scales need gradient updates.

2. Activation Clone: During training, a loss matches a wide slice of intermediate activations (attention outputs, feed-forward activations) between teacher and student after projection wiht RME. These alignment losses ensure the behaviour of the compressed student remains faithful to the teacher despite the drastic parameter reduction.

Experiments using strong open-source teachers (Llama-3.2-3B, Qwen-2.5-3B/7B) show that 1.5-4 B-parameter students trained on 10–20 B tokens reach or exceed the accuracy of state-of-the-art SLMs that were pre-trained on trillions of tokens, amounting to ≈ 1000× training-token efficiency. Extensive ablations confirm the importance of FFN alignment and the built-in alignment-free property of the projections.

**Questions:**

1. What do you see as the main bottlenecks for LRC’s few-shot or in-context learning ability, and might injecting short instruction-tuning episodes during pre-training help close that gap?
2. Given the mixed results on ToxiGen/TruthfulQA, do you envisage incorporating refusal demonstrations or lightweight RLHF after LRC without negating the token-efficiency advantage?

**Ethical Concerns:**

["NO or VERY MINOR ethics concerns only"]

**Final Justification:**

My concerns are addressed and I will boost my significance score to 3, retaining the main score.

**Limitations:**

The authors include a candid “Limitations and Broader Impact” section that notes untested large-scale regimes, absence of RLHF, and potential weaknesses on maths/coding and safety-critical use-cases. This openness is appreciated; adding quantitative resource comparisons and a discussion of cross-lingual transfer would strengthen it further.

**Quality:**

3

**Strengths And Weaknesses:**

### Strengths

* Interesting idea about using the projections as the only tunable parameters combined with efficient training method make this setup appealing.
* Authors demonstrate that FFN activations are more critical than attention maps for distillation, and that high-quality data can offset smaller budgets.
* The method achieves competitive downstream accuracy with 10–20 B pre-training tokens versus ≥ 1 T for baselines.
* Benchmarks span reasoning, commonsense, comprehension, and knowledge tasks; ablations dissect projection, clone loss terms, data quality, weight sharing, and α sensitivity.
* Training throughput on H800 GPUs, memory analyses, and pseudo-code facilitate reproduction; open-source code link provided.

### Weaknesses

* LRC underperforms Qwen3/Gemma in 5-shot settings (Table 15), suggesting limited in-context learning despite strong zero-shot scores.
* Results lack confidence intervals or multiple runs; authors cite cost constraints but some variability analysis would strengthen claims.
* Post-training on UltraChat offers limited gains; negative impact on IFeval indicates that LRC alone is insufficient for instruction following and safety.

---

> ### Author Rebuttal · Authors · 2025-07-31
>
> Thank you for your very positive and encouraging review. We are delighted that you found our core idea *"interesting"* and our training setup *"appealing."* We especially appreciate you highlighting the strength of our experimental validation and the clarity of our presentation. Your questions have given us a valuable opportunity to elaborate on the nuances of our method.
>
> We address your questions and weaknesses below.
>
> ---
>
> > **W1, Q1: The performance of LRC models in few-shot settings is limited. What do you see as the main bottlenecks for LRC’s few-shot or in-context learning ability?**
>
> Thank you for highlighting this important point. We agree that few-shot performance is a critical capability for modern language models, and we appreciate your suggestion regarding the potential cause.
>
> Our hypothesis aligns with your intuition: the main bottleneck in LRC’s few-shot or in-context learning ability stems from the limited exposure to long-context and instruction-tuning data during training. These components are essential for cultivating strong in-context reasoning and task generalization.
>
> To verify this, we conducted a new experiment where we continued training our **LRC-4B** model on an additional 0.5B tokens of long-context data (up to 8K sequence length) from Fineweb-edu and instruction data from OpenHermes. The results are shown in Table R1 below.
>
> ***Table R1:** 5-shot performance on various benchmarks with/without extra long-context and instruction-tuning data. (Last three rows are taken from Table 15 of the main paper.)*
>
> |Benchmark|WinoGrande|ARC-C|BoolQ|MMLU|Avg.↑|
> |:--|:--|:--|:--|:--|:--|
> |**LRC-4B-Long (new)**|71.19|59.39|86.45|65.52|70.64|
> |LRC-4B (original)|69.93|58.36|85.69|65.10|69.77|
> |Qwen3-4B|66.85|61.18|85.27|70.04|70.84|
> |Minitron-4B|73.95|53.58|82.39|57.86|66.95|
> |Gemma3-4B|69.06|60.49|84.77|58.33|68.16|
>
> The newly trained **LRC-4B-Long** model achieves a **+1.0%-point average improvement**, with consistent gains across all benchmarks. This supports our hypothesis that the earlier performance limitation was not due to LRC's architecture or methodology, but rather the composition of its training data. While LRC-4B-Long does not yet surpass Qwen3-4B on average (likely due to Qwen’s extensive proprietary instruction-tuning and larger training corpus), it outperforms strong open baselines like Minitron-4B and Gemma3-4B, showcasing LRC’s potential when equipped with richer training signals.
>
>
>
> > **W2: Results lack confidence intervals or multiple runs.**
>
> We agree that evaluating variability is crucial for demonstrating the robustness and reliability of our method.
>
> While the high computational cost of pre-training limits our ability to perform extensive multi-run experiments across all settings, we conducted a second run of our key LRC-1.5B experiment using a different random seed (42) to assess result stability. The performance across both runs is presented in Table R2.
>
> ***Table R2:** Stability of LRC under different random seeds.*
>
> | Benchmark | ARC-E | ARC-C | LogiQA | CSQA | PIQA | WinoG | BoolQ | SciQ | MMLU | Avg.↑|
> | :--- | :--- | :--- | :--- | :--- | :--- | :--- | :--- | :--- | :--- | :--- |
> | LRC-1.5B (Seed 218) | 74.75 | 44.97 | 30.72 | 65.77 | 73.07 | 62.25 | 75.78 | 94.60 | 49.42 | 63.48 |
> | **LRC-1.5B** (Seed 42) | 76.81 | 42.75 | 30.72 | 63.64 | 72.31 | 62.43 | 77.86 | 94.10 | 49.05 | 63.30 |
> | MiniCPM-1.2B | 70.16 | 39.68 | 30.88 | 64.29 | 74.65 | 60.77 | 67.58 | 91.50 | 44.23 | 60.42 |
>
> The performance variance between the two runs is minimal (<0.2). This result increases our confidence in the robustness of LRC. Due to time constraints, we can only run one more for now. We will conduct additional runs later to provide more comprehensive mean and variance statistics.
>
>
> > **W3: Post-training on UltraChat offers limited gains; negative impact on IFeval.**
>
> Thank you for this observation. The UltraChat dataset, while helpful for standard instruction-following benchmarks, is **out-of-distribution** for the complex, multi-turn instructions found in IFeval. As a result, SFT on UltraChat can inadvertently bias the model toward generic instruction styles, leading to decreased performance on more intricate tasks.
>
> This reinforces an important takeaway: LRC, as a pre-training method, is not a substitute for carefully curated alignment data. Instead, it provides a strong foundation that benefits greatly from targeted post-training with diverse and challenging instruction datasets. In future work, we plan to investigate training with more instructionally diverse corpora to mitigate distributional shifts and improve generalization to harder instruction-following benchmarks like IFeval.
>
> > **Q2: Given the mixed results on ToxiGen/TruthfulQA, do you envisage incorporating refusal demonstrations or lightweight RLHF?**
>
> Absolutely. LRC is fully compatible with post-training alignment techniques like RLHF. The goal of LRC is to create a highly capable and knowledge-rich base model with maximum token efficiency. This efficient base model can then serve as a superior starting point for standard alignment procedures.
>
> Furthermore, LRC's ability to transfer behavioral patterns opens up interesting possibilities. Since LRC can generalize from seen data (e.g., training on "The highest mountain is Everest" might co-activate "The world's second highest peak is Chhogori"), one could potentially apply the LRC framework during an RLHF-like process to allow the student to directly inherit the safety preferences and alignment of a well-tuned teacher model. This could make the alignment process itself more efficient.
>
> Therefore, incorporating refusal demonstrations or a lightweight RLHF phase is the natural and necessary next step. Doing so would not negate LRC's efficiency advantage; rather, it would leverage it by reducing the cost of the most expensive phase (pre-training) and allowing resources to be focused on the critical alignment stage. We see this as a promising direction for future work.
>
> ---
>
> We are sincerely grateful for your insightful and detailed feedback. In response to the points you raised, we have conducted additional experiments and incorporated new analyses. We hope that these additions address your concerns and will be included in the revised manuscript. Please do not hesitate to let us know if you have any further questions.

---

> > ### Comment · Reviewer_sUgd · 2025-08-06
> >
> > Thank you for the thoughtful rebuttal. My concerns are addressed and I will boost my significance score to 3, retaining the main score.
> >
> > The extra long-context/instruction run and the second seed both help substantiate LRC’s stability and its headroom on few-shot tasks; the +1 pp average improvement is encouraging and the low run-to-run variance (<0.2) alleviates my earlier robustness concern. I also appreciate the nuanced discussion of UltraChat drift and the roadmap for post-LRC alignment.

---

> > > ### Author Response · Authors · 2025-08-06
> > >
> > > Thank you for your response. We are pleased that your earlier concerns have been resolved. If you have any additional thoughts or questions, please do not hesitate to let us know. We would be happy to provide any further clarification you may need.
> > >
> > > We sincerely appreciate your valuable feedback and encouragement!

---

### Official Review · Reviewer_qYJs · 2025-07-03

**Clarity:** 3
**Significance:** 3
**Originality:** 3
**Rating:** 4
**Confidence:** 3

**Summary:**

This paper introduces Low-Rank Clone (LRC), an efficient pre-training method for constructing Small Language Models (SLMs) from larger teacher models via joint soft pruning and knowledge distillation. LRC leverages trainable low-rank projection matrices to generate the student model’s weights directly from the teacher, while simultaneously aligning internal activations—particularly the often-overlooked Feed-Forward Network (FFN) activations—to preserve behavioral equivalence.

**Questions:**

1. Can LRC scale efficiently to larger models (e.g., 13B, 30B)? Experiments or insights on memory and performance tradeoffs at larger scales would strengthen the significance of the paper.
2. The ablations highlight FFN signals as especially valuable. Can the authors elaborate on why FFNs carry more transferable knowledge than attention layers?
3. The learnable projection matrices are central to LRC. After training, what structural properties do these matrices exhibit? For instance, do they learn to selectively prune or transform specific types of information from the teacher's weight matrices? An analysis of the learned projections could provide deeper insights into how knowledge is being transferred.
4. The paper shows compatibility with 8-bit quantization. How does LRC interact with other compression methods? For example, could a student model generated by LRC be further pruned using traditional structured or unstructured pruning techniques for an additional boost of efficiency?

**Ethical Concerns:**

["NO or VERY MINOR ethics concerns only"]

**Final Justification:**

I've carefully considered the authors' rebuttal, in which detailed clarifications and extra experiments are provided and address my concerns. Provided that these clarifications and extra experiments are included in the revised version of the paper, I think this work is acceptable. Thus I've upgraded my original score.

**Limitations:**

yes

**Quality:**

3

**Strengths And Weaknesses:**

Summary of Strengths:
1. The paper presents a technically sound and well-engineered method that significantly improves training efficiency without sacrificing performance.
2. Extensive experiments are performed, including comparison with strong baselines, comprehensive ablations, and performance scaling studies.
3. Training SLMs efficiently has become increasingly important as compute costs rise. LRC offers a compelling solution with potential real-world applications in resource-constrained environments.

Summary of Weaknesses:
1. The paper focuses mainly on small to mid-sized SLMs.  It remains unclear how LRC scales beyond 7B parameters or with more extensive pre-training.
2. As the authors acknowledge in the appendix, the performance of LRC models in few-shot settings is less impressive than their zero-shot capabilities, and they lag behind some baselines. While the paper offers plausible hypotheses for this (e.g., lack of long-context data, less extensive SFT), this is a notable limitation, as in-context learning is a critical capability for large language models. This weakness suggests that while LRC is excellent at transferring static knowledge, it may be less effective in transferring the ability for rapid, on-the-fly adaptation.
3. The paper demonstrates LRC's effectiveness on SLMs up to 4B parameters. While impressive, it remains an open question how this method scales to larger models or if there is a point where the low-rank assumption becomes a bottleneck, preventing the student from capturing the full complexity of a much larger teacher (e.g., a 70B or larger model). The limitations section touches on this, but I think this should be investigated experimentally and analytically in the paper to ensure a better significance and soundness of the paper.

---

> ### Author Rebuttal · Authors · 2025-07-31
>
> Thank you for your insightful and constructive review. We sincerely appreciate your recognition as *"technically sound, well-engineered"* and *"extensive experiments performed."* Your feedback has been invaluable in helping us strengthen the paper.
>
> Below, we address your concerns in detail.
>
> ---
>
> > **W1, W3, Q1: Scalability of LRC (experimentally/analytically)? Would the low-rank assumption become a bottleneck?**
>
> We believe LRC is scalable and may become even more advantageous as model size increases.
>
> **Analytical Scalability:** The scalability of LRC is supported by the **Johnson-Lindenstrauss (JL) Lemma**, which explains why our low-rank projection becomes more effective for larger models:
> - **Formulation:** LRC compresses a teacher’s weight matrix $\mathbf{W}^T$ via a projection $\mathbf{W}^S = \mathbf{W}^T \mathbf{W}^p$, where each row of $\mathbf{W}^T$ ($n=d_\text{ffn}^T$) is a point in $d_{\text{model}}^T$-dimensional space. LRC aims to preserve the geometry of these points using activation clone (we have explained this in detail in our responses to Q2 and Q3).
> - **JL Lemma:** It guarantees that this geometry can be preserved if the projected dimension $d^S$ satisfies $d^S\ge O(\log d_\text{ffn}^T/\varepsilon^2)$.
> - **Implication for LRC:** As model size increases from 3B to 70B, $d_\text{ffn}^T$ increases moderately (e.g., from 11k to 29k), while the theoretically required $d^S$ grows only logarithmically. This provides a much larger "dimensional budget" for our low-rank projection at larger scales, making it easier to find a high-fidelity projection.
>
> **Experimental Evidence:** To further validate scalability, we launched a new experiment, training a **LRC-7B from a larger teacher Qwen2.5-14B-Instruct** for 20B tokens. While this experiment is ongoing due to resource constraints, we present the intermediate MMLU scores in Table R1.
>
> ***Table R1:** The trend of MMLU scores with increasing training tokens.*
>
> |Train. Progress (tokens)|2B|4B|6B|8B|
> |--|--|--|--|--|
> |LRC-4B|48.98|58.08|60.55|61.81|
> |**LRC-7B**|51.29|61.11|64.58|67.16|
>
> These results clearly show that LRC exhibits strong scaling properties. The new LRC-7B model consistently outperforms the LRC-4B model at every checkpoint, and its performance **continues to rise steadily**. This trend suggests that LRC's effectiveness is not limited to smaller models.
>
> > **W2: The performance of LRC models in few-shot settings is less impressive.**
>
> We believe the limited few-shot performance primarily stems from the lack of long-context and instruction-tuning data, both crucial for in-context learning. To investigate, we further fine-tune **LRC-4B** on an additional 0.5B tokens comprising long-context (up to 8K) and instruction-tuning data sampled from Fineweb-edu and OpenHermes.
>
> ***Table R2:** 5-shot performance on various benchmarks with/without extra long-context and instruction-tuning data. (Last three rows are taken from Table 15 of the main paper.)*
>
> |Benchmark|WinoGrande|ARC-C|BoolQ|MMLU|Avg.↑|
> |:--|:--|:--|:--|:--|:--|
> |**LRC-4B-Long (new)**|71.19|59.39|86.45|65.52|70.64|
> |LRC-4B (original)|69.93|58.36|85.69|65.10|69.77|
> |Qwen3-4B|66.85|61.18|85.27|70.04|70.84|
> |Gemma3-4B|69.06|60.49|84.77|58.33|68.16|
>
> As shown in Table R2, the enhanced **LRC-4B-Long** achieves a **+1.0% average improvement** over the **original LRC-4B**, indicating that few-shot capability can be significantly enhanced with higher-quality, long-context training data.
>
> While the enhanced LRC-4B model still trails Qwen3-4B, likely due to differences in data scale and tuning strategies, it outperforms other competitive baselines like Minitron-4B (see Table 15) and Gemma3-4B. This confirms that the limitation is the quality of the current training data.
>
> > **Q2: Why do FFNs carry more transferable knowledge than attention?**
>
> Our core hypothesis is that FFNs and attention layers capture different types of knowledge. FFNs primarily store **factual and world knowledge** within their parameters [1, 2], while attention mechanisms focus more on capturing **token-level, contextual relationships**, such as syntax and co-reference. Therefore, transferring the knowledge embedded in FFNs is especially critical for equipping the student model with foundational understanding.
>
> This view is supported by prior work [1, 2], which identifies FFNs as key-value memories storing knowledge from the pre-training corpus. Specifically, the FFN activation, $\text{act}=\text{silu}(x\mathbf{W}\_{\text{gate}}) \cdot x\mathbf{W}\_{\text{up}}$, measures the similarity between $x$ and the knowledge stored in FFN. Our activation clone forces replicating this similarity distribution, driving the low-rank projection to project the teacher weights into a similar distribution.
>
> Our ablation results in Figure 3 empirically support this view: removing the FFN clone loss (`LRC w/o FFN`) leads to a significant and persistent drop in performance, while the model recovers more easily from removing the attention clone loss (`LRC w/o Attn`).
>
> To provide more direct evidence, we conducted a **new neuron-masking experiment** on a factual QA task (e.g., "Who was the first emperor of ancient Rome?" → "Augustus"):
>
> 1. We input factual questions to the teacher model and identified the top 50 FFN neurons with high activations, termed "important neurons."
> 2. We masked the **same neuron indices** in the student model's FFNs.
> 3. As a baseline, we masked 50 random neurons in the student model.
>
> ***Table R3:** Performance of different neuron-masking methods (teacher: Llama-3.2-3B, student: LRC-1.5B).*
>
> |Score Type|Teacher|Student|
> |:---|:---|:---|
> |Original Score|0.85|0.48|
> |**Important Neurons Masked**|**0.62 (-27%)**|**0.33 (-31%)**|
> |Random Neurons Masked|0.85|0.49|
>
> As shown in Table R3, masking the important neurons causes significant performance degradation in both teacher and student, while random masking has minimal impact. These results confirm that:
>
> - FFNs encode factual knowledge in FFN neurons;
> - **LRC effectively transfers this knowledge by aligning the student's activation patterns with teacher's.**
>
> > **Q3: What structural properties do these matrices exhibit? Do they learn to selectively prune or transform specific types of information?**
>
> Our hypothesis is that for general-purpose distillation, the low-rank projection in LRC are primarily **structure-preserving**, aiming to retain the rich and diverse capabilities of the teacher model rather than selectively pruning information. This is essential for cloning a broad range of knowledge encoded in the teacher's parameters.
>
> To investigate this, we conducted two complementary analyses:
>
> 1. **SVD Analysis:**
> We tracked the singular values of a representative projection matrix, $\mathbf{W}^p_{\text{up}}$, during training. As shown in Table R4, the singular values increase across training, suggesting that the projection matrix retains high-rank structure and does not collapse into a small subspace. This indicates that the projection actively utilizes the full dimensionality of the teacher, supporting the goal of comprehensive knowledge transfer.
>
>     ***Table R4:** Evolution of singular values during training.*
>     |Singular Rank Percentage|0%|10%|...|50%|...|90%|100%|
>     |:--|:--|:--|:--|:--|:--|:--|:--|
>     |Train. 10%|6.14|2.48|...|1.72|...|0.79|0.37|
>     |Train. 50%|6.69|3.62|...|2.74|...|1.61|0.42|
>     |Train. 100%|6.70|3.85|...|2.88|...|1.67|0.38|
> 2. **Structural Similarity Analysis:**
> We evaluated whether LRC preserves the internal weight geometry of the teacher by comparing similarity matrices $\text{Sim}=\mathbf{W}\mathbf{W}^\top$ of the FFN weights. Specifically, we computed the MSE between the teacher and student similarity matrices, and compared this to a baseline with randomly initialized weights. As shown in Table R5, LRC significantly reduces the structural discrepancy compared to the random baseline, demonstrating that the student retains internal structural patterns of the teacher.
>
>     ***Table R5:** Structural Similarity of FFN Weights.*
>     |Exp. Type|MSE (Teacher vs. Student)|MSE (Teacher vs. Random)|
>     |:--|:--|:--|
>     |$\mathbf{W}_{\text{up}}$|**0.000576**|0.001026|
>     |$\mathbf{W}_{\text{gate}}$|**0.000635**|0.001202|
>     |$\mathbf{W}_{\text{down}}$|**0.000559**|0.001124|
>
> While we believe selective pruning is less likely for general distillation, we see this as an exciting direction for future work, where LRC could be extended to specialize in specific tasks or domains.
>
> > **Q4: How does LRC interact with other compression methods like pruning?**
>
> LRC is fully compatible with other compression techniques, including structured pruning. To demonstrate this, we applied LLM-Pruner [3] to our LRC-1.5B model, pruning 20% of its parameters. We then conducted a brief 2-hour LoRA fine-tuning to restore performance, resulting in LRC-Pruned-1.2B.
>
> ***Table R6:** Performance of LRC combined with LLM-Pruner.*
> |Benchmark|ARC-E|ARC-C|LogiQA|CSQA|PIQA|WinoG|BoolQ|SciQ|MMLU|Avg.↑|
> |:--|:--|:--|:--|:--|:--|:--|:--|:--|:--|:--|
> |LRC-1.5B|74.75|44.97|30.72|65.77|73.07|62.25|75.78|94.60|49.42|63.48|
> |**LRC-Pruned-1.2B**|71.93|40.44|29.34|61.02|71.44|58.01|75.11|93.40|44.66|60.59|
> |MiniCPM-1.2B|70.16|39.68|30.88|64.29|74.65|60.77|67.58|91.50|44.23|60.42|
>
> The pruned model outperforms the strong **MiniCPM-1.2B** baseline across most benchmarks as shown in Table R6. This confirms that *LRC can be seamlessly integrated with structured pruning techniques* to produce even smaller and more efficient models.
>
> ---
>
> Thank you once again for your constructive feedback. We hope that the new experiments and analyses have addressed your concerns. We will add these additional results to the revised paper. If you have any further questions, please feel free to ask.
>
> [1] Transformer Feed-Forward Layers Are Key-Value Memories.
>
> [2] Locating and Editing Factual Associations in GPT.
>
> [3] LLM-Pruner: On the Structural Pruning of Large Language Models.

---

> ### Author Response · Authors · 2025-08-05
>
> Dear reviewer qYJs,
>
> Thank you for taking the time to provide us with your thoughtful and constructive feedback. We greatly appreciate your insights and have prepared our rebuttal to address each of your concerns with the utmost respect.
>
> We have updated Table R1 as LRC-7B continues training up to 12B tokens. As shown in udpated Table R1 below, LRC exhibits **strong scalability**. The new LRC-7B model consistently outperforms the LRC-4B model at every checkpoint, and its performance continues to improve steadily. This trend indicates that the effectiveness of LRC is not limited to small-scale models.
>
> ***Table R1 (updated):** The trend of MMLU scores with increasing training tokens.*
>
> |Train. Progress (tokens)|2B|4B|6B|8B|10B|12B|
> |--|--|--|--|--|--|--|
> |LRC-4B|48.98|58.08|60.55|61.81|62.11|63.45|
> |**LRC-7B**|51.29|61.11|64.58|67.16|67.91|69.12|
>
> We kindly ask if you could confirm whether our responses have sufficiently resolved your concerns. Please feel free to reach out with any further questions or suggestions, as we remain open to continued discussion.
>
> Best regards,
> The Authors

---

> > ### Comment · Reviewer_qYJs · 2025-08-07
> >
> > Thank the authors for the clarifications and additional results. I will raise my score accordingly while strongly requesting that these clarifications and additional results be incorporated into the next version of the paper.

---

> > > ### Author Response · Authors · 2025-08-07
> > >
> > > Thank you for your response. We are pleased that your concerns have been resolved. We will add all the analyses and experiments supplemented during the rebuttal period to the revised version of the paper. If you have any further questions, we would be very happy to provide any further clarification you may need.
> > >
> > > We sincerely appreciate your valuable feedback and encouragement!

---

### Note · Authors · 2025-08-11

We are grateful to the reviewers for their insightful feedback, which has significantly strengthened our paper.

We are delighted that all reviewers recognized the core strengths of our work: its novelty as a "fresh idea that is potentially impactful" (Rev 4MPA), its "technically sound and well-engineered" method (Rev qYJs), its "appealing" training efficiency (Rev sUgd), and its "thorough" experimental validation (Rev Vtru).

The reviewers raised excellent questions regarding scalability, few-shot performance, architectural constraints, and comparisons to state-of-the-art baselines. We addressed these points comprehensively through a series of new experiments conducted during the rebuttal period:

1. Scalability and Performance: We demonstrated strong scaling trends with a new LRC-7B model, addressing Rev qYJs's concerns.
2. Few-shot Learning: We showed that few-shot performance is primarily data-dependent by significantly improving results with additional long-context and instruction-tuning data, resolving a key weakness noted by Rev qYJs and Rev sUgd.
3. Efficiency Comparison: As requested by Rev 4MPA, we provided a direct comparison against a Minitron baseline, showing LRC's superior convergence and final performance.
4. Compatibility: We demonstrated LRC's compatibility with structured pruning (LLM-Pruner) to show that architectural constraints (e.g., FFN size) are not fundamental limitations, addressing concerns from Rev 4MPA.
5. Ablations & Analysis: We provided new ablation studies on the KL loss term (Rev Vtru) and neuron-masking experiments (Rev qYJs, 4MPA, Vtru) to further validate the critical role of FFN activation cloning.

We are pleased that these extensive additions successfully resolved all questions. All four reviewers have explicitly acknowledged that their concerns were addressed.

We will incorporate all new results and analyses into the final manuscript. Thank you for your time and consideration.

---

### Decision · Program_Chairs · 2025-09-17

**Decision:**

Accept (spotlight)

**Comment:**

(a) Summary

This paper introduces Low-Rank Clone (LRC), an efficient pre-training method for constructing Small Language Models (SLMs) from larger teacher models via joint soft pruning and knowledge distillation. LRC leverages trainable low-rank projection matrices to generate the student model’s weights directly from the teacher, while simultaneously aligning internal activations—particularly the often-overlooked Feed-Forward Network (FFN) activations—to preserve behavioral equivalence.

Low-Rank Projection: For every transformer layer, the teacher’s large weight matrices are factorised through learnable low-rank projectors; the projected (compressed) matrices are used directly as the student’s parameters, so the student is generated on-the-fly rather than trained from scratch. Only the projector weights and RMSNorm scales need gradient updates.

Activation Clone: During training, a loss matches a wide slice of intermediate activations (attention outputs, feed-forward activations) between teacher and student after projection wiht RME. These alignment losses ensure the behaviour of the compressed student remains faithful to the teacher despite the drastic parameter reduction.

Experiments using strong open-source teachers (Llama-3.2-3B, Qwen-2.5-3B/7B) show that 1.5-4 B-parameter students trained on 10–20 B tokens reach or exceed the accuracy of state-of-the-art SLMs that were pre-trained on trillions of tokens, amounting to ≈ 1000× training-token efficiency. Extensive ablations confirm the importance of FFN alignment and the built-in alignment-free property of the projections.
The paper shows that this algorithm is much more token-efficient than standard pretraining from scratch and more time-efficient than TinyBert (distillation with no pruning).

(b) Strengths

The paper presents a technically sound and well-engineered method that significantly improves training efficiency without sacrificing performance.

Authors demonstrate that FFN activations are more critical than attention maps for distillation, and that high-quality data can offset smaller budgets. The main idea of this work, that is, using trainable projections for LLM shrinking, is novel and seems very promising. These projections introduce token-independent overhead and can be multiplied out after training, making this inference efficient. Since teacher weights are frozen, this potentially allows the student to not forget information from the teacher so easily at the beginning, in comparison to initialization with pruning, allowing the student to benefit from it longer. The idea sounds simple and intuitive, yet it is not present in the literature according to my knowledge, strengthening the potential impact.
This method can be used alongside distillation and makes hidden state matching more efficient, as these projections can be reused to align activations.

Extensive experiments are performed, including comparison with strong baselines, comprehensive ablations, and performance scaling studies. The method achieves competitive downstream accuracy with 10–20 B pre-training tokens versus ≥ 1 T for baselines.
Benchmarks span reasoning, commonsense, comprehension, and knowledge tasks; ablations dissect projection, clone loss terms, data quality, weight sharing, and α sensitivity. Training throughput on H800 GPUs, memory analyses, and pseudo-code facilitate reproduction; open-source code link provided.

(c) Weaknesses

This paper introduces a novel, promising soft-shrinking technique that appears to be a good alternative to the dominant shrinking + distillation recipe. However, the initial submission did not contain a proper validation confirming the validity of the method, comparing their results to distillation only (while the low-rank clone model was also distilled). Although the authors submitted this during the rebuttal period, this strongly changes the conclusions of the work, as a low-rank clone is only superior for long horizons, which are not always practical for shrinking. Moreover, the paper did not mention that this method does not decrease the width of feed-forward and attention head size, which may become problematic for hierarchical shrinking, which is the most practical application.

(d) The most important reasons for your decision to accept/reject.

The effectiveness and novelty of the approach.

(e) Summarize the discussion and changes during the rebuttal period.

The rebuttal was extensive and satisfied most of the reviewers' concerns, leading to increase in the rating for one reviewer (from 3 to 4), with two others maintaining their original high score of 5.